



# New Model of Reactive Transport in Single-Well Injection-Withdrawal Test with Aquitard Effect

Quanrong Wang[1]*, Wenguang Shi[1], and Hongbin Zhan[2]*

[1]School of Environmental Studies, China University of Geosciences, Wuhan, Hubei, 430074, P. R. China
[2]Department of Geology and Geophysics, Texas A& M University, College Station, TX 77843-3115, USA

*Correspondence to*: Quanrong Wang (wangqr@cug.edu.cn), and Hongbin Zhan (zhan@geos.tamu.edu)

**Abstract.**

The model of single-well injection-withdrawal (SWIW) test has been widely used to investigate reactive radial dispersion in remediation or parameter estimation of the *in situ* aquifers. Previous analytical solutions only focused on a completely
isolated aquifer for the SWIW test, excluding any influence of aquitards bounding the tested aquifer. This simplification might be questionable in field applications when test durations are relatively long, because solute transport in or out of the bounding aquitards is inevitable due to molecular diffusion and cross-formational advective transport. Here, a new SWIW model is developed in an aquifer-aquitard system, and the analytical solution in the Laplace domain is derived. Four phases of the test are included: the injection phase, the chaser phase, the rest phase and the extraction phase. The Green's function
method is employed for the solution in the extraction phase. As the permeability of aquitard is much smaller than the permeability of the aquifer, the flow is assumed to be perpendicular to the aquitard, thus only vertical dispersive and advective transports are considered for aquitard. The validity of this treatment is tested by a numerical solution. The sensitivity analysis demonstrates that the influence of vertical flow velocity and porosity in the aquitards, and radial dispersion of the aquifer is more sensitive to the SWIW test than other parameters. In the injection phase, the larger radial
dispersivity of the aquifer could result in the smaller values of breakthrough curves (BTCs), while greater values of BTCs of the chaser and rest phases. In the extraction phase, it could lead to the smaller peak values of BTCs. The new model of this study performs better than previous studies excluding the aquitard effect for interpreting data of the field SWIW test.

**Keywords**: Aquifer-aquitard system; Radial dispersion; Parameter estimation; Push-pull test



## 1 Introduction

A single-well injection-withdrawal (SWIW) test could be applied for investigating aquifer properties related to reactive transport in subsurface instead of the inter-well tracer test, due to its advantages of efficiency, low cost, and easy implementation. The SWIW test is sometimes called the single-well push-pull test, or single-well huff-puff test, or single-well injection-backflow test (Jung and Pruess, 2012). A complete SWIW test includes the injection, the chaser, the rest, and the extraction phase. The second and third phases are generally ignored in the analytical solutions, but recommended in the field applications, since they could increase the reaction time for the injected chemicals with the porous media (Phanikumar and McGuire, 2010;Wang and Zhan, 2019).

Similar to other aquifer tests, the SWIW test is a forced-gradient groundwater tracer test, and analytical solutions are often preferred to determine the *in situ* aquifer properties, due to the computational efficiency. Currently, many analytical models were available for various scenarios of the SWIW tests (Gelhar and Collins, 1971; Huang et al., 2010; Chen et al., 2017; Schroth and Istok, 2005; Wang et al., 2018). However, these studies were based on a common underlying assumption, that the studied aquifer was isolated from adjacent aquitards. In another word, the aquitards bounding the aquifer are taken as two completely impermeable barriers for solute transport. To date, numerous studies demonstrated that such an assumption might cause errors for groundwater flow (Zlotnik and Zhan, 2005;Hantush, 1967), and for reactive transport (Zhan et al., 2009; Chowdhury et al., 2017; Li et al., 2019). This is because even without any flow in the aquitards, molecular diffusion is inevitable to occur when solute injected to the aquifer is close to the aquitard-aquifer interface. This is particularly true when a fully penetrating well is used for injection, thus a portion of injected solute is very close to the aquitard-aquifer interface and the SWIW test duration is relatively long so the effect of molecular diffusion can be materialized. Another important point to note is that the materials of aquitard are usually clay and silt which have strong absorbing capability for chemicals and great mass storage capacities (Chowdhury et al., 2017). Actually, the influence of aquitard on reactive transport in aquifers has attracted attentions for several decades. As for radial dispersion, Chen (1985), Wang and Zhan (2013) and Zhou et al. (2017) presented analytical solutions for radial dispersion around an injection well in an aquifer-aquitard system. However, these models only focus on the first phase of the SWIW test (injection).

Another assumption included in many previous models of radial dispersion is that the concentration of the mixing water with the injected tracer is equal to the injected tracer concentration during the injection phase. The examples of employing such an assumption include Gelhar and Collins (1971), Chen (1985, 1987), Moench (1989), Chen et al. (2007, 2012), Schroth et al. (2001), Tang and Babu (1979), Chen et al. (2017), Huang et al. (2010), Chen et al. (2012), and Zhou et al. (2017). This assumption implies that the mixing effect in the wellbore is not considered, where the mixing effect refers to the mixture between the original (or native) water and the injected tracer in the well. Such effect is excluded in almost all previous studies except Wang et al. (2018) for the SWIW test, who developed two-phase (injection and extraction) models with specific considerations of the mixing effect. In many field applications, the chaser and rest phases are generally involved and the mixing effect also happens in these two phases in the SWIW test.



Besides above-mentioned issues in previous studies, another issue is that the advection-dispersion equation (ADE) was used to govern the reactive transport of SWIW tests (Gelhar and Collins,1971; Wang et al.; 2018; Jung and Pruess, 2012). The

validity of ADE was challenged by numerous laboratory and field experimental studies in heterogeneous media, mostly because ADE could not adequately interpret anomalous reactive transport, e.g. the early arrivals and/or heavy late-time tails of the breakthrough curves (BTCs). Alternatively, the multi-rate mass transfer (MMT) model was proposed to interpret the data of SWIW test (Huang et al., 2010; Chen et al., 2017). In the MMT model, the porous media is divided into two domains: One mobile domain where both dispersion and advection happen, and other immobile domain that only diffusion occurs

(Haggerty et al., 2000;Haggerty and Gorelick, 1995). A subset of MMT is the mobile-immobile model (MIM) in which the mass transfer between two domains becomes a single parameter instead of a function. The MIM model can grasp most characteristics of MMT and is mathematically simpler than MMT. Besides the MMT model, the continuous time random walk (CTRW) model and the fractional advection-dispersion equation (FADE) model were also applied for anomalous reactive transport in SWIW tests (Hansen et al., 2017; Chen et al., 2017). However, due to the complexity of the mathematic

models of CTRW and FADE, it is very difficult, or even not possible to derive analytical solutions for those two models, although both methods perform well in a numerical framework.

In this study, a new model of SWIW tests will be established by including both mixing effect in the wellbore and the aquitard effect under the MIM framework. The reason to choose MIM as the working framework is to capture the possible anonymous transport characteristics that cannot be described by ADE but at the same time to make the analytical treatment

of the problem possible. Four stages of a SWIW test will be considered. The model of the mixing effect will be developed using a mass balance principle in the chaser and rest phases. Analytical solution will be derived to facilitate the data interpretation for SWIW tests. The newly developed model will be checked against numerical solutions and field experimental data.

## 2 Model statement of the SWIW test

A single test well is assumed to fully penetrate an aquifer with uniform thickness. Both the aquifer and aquitards are homogeneous and extend laterally to infinity. The concept of homogeneity here deserves some clarification. First, despite the fact that the homogeneity assumption is commonly used in developing analytical models of subsurface flow and transport, one should be aware that a rigorous sense of homogeneity probably never exists in a real-world setting (unless the media are composed of idealized glass balls as in some laboratory experiments). Therefore, the homogeneity concept here should be

envisaged as a media whose hydraulic parameters vary within relatively narrow ranges, or the so-called weak heterogeneity. Some examples of weak heterogeneity include the Borden Site of Canada (Sudicky, 1988). Wang et al. (2018) employed a stochastic modeling technique to test the assumption of homogeneity associated with the SWIW test, and found that such an assumption could be used to approximate a heterogeneous aquifer when the variance of spatial hydraulic conductivity was small. Second, for moderate or even strong heterogeneous media such as Cape Code site (Hess, 1989) or MADE site



(Bohling et al., 2012), the analytical model developed under the homogeneity assumption is also valuable, but in a statistical sense, as long as the media heterogeneity can be regarded as spatially stationary, meaning that the statistical structure of the media heterogeneity does not vary in space. In this setting, the analytical model developed under the homogeneity assumption is used to describe the (ensemble) average characteristics of an ensemble of heterogeneous media which are statistically identical but individually different. In another word, such an analytical model will provide a statistically average

description of many realizations (an ensemble) which are similar to the heterogeneous media of concern, but it cannot provide an exact description for the particular heterogeneous media under investigation.

A cylindrical coordinate system is employed in this study, and the origin is located at the well center. The *z*-axis and the *r*-axis are vertical and horizontal, respectively. A schematic diagram of the model investigated by this study is similar to Figure 1 of Wang and Zhan (2013).

## 100  2.1 Reactive transport model

Considering advective effect, dispersive effect and first-order chemical reaction in describing solute transport under the MIM framework, the governing equations the SWIW test are:

$$\theta_m R_m \frac{\partial C_m}{\partial t} = \frac{\theta_m}{r} \frac{\partial}{\partial r}\left(r D_r \frac{\partial C_m}{\partial r}\right) \mp \theta_m v_a \frac{\partial C_m}{\partial r} - \omega_a(C_m - C_{im}) - \theta_m \mu_m C_m - \left(\pm \frac{\theta_{um} v_{um}}{2B} C_{um} - \frac{\theta_{um} D_u}{2B} \frac{\partial C_{um}}{\partial z}\right)\Big|_{z=B} +$$

$$\left(\mp \frac{\theta_{lm} v_{lm}}{2B} C_{lm} - \frac{\theta_{lm} D_l}{2B} \frac{\partial C_{lm}}{\partial z}\right)\Big|_{z=-B}, r \geq r_w, \tag{1a}$$

$$\theta_{im} R_{im} \frac{\partial C_{im}}{\partial t} = \omega_a(C_m - C_{im}) - \theta_{im} \mu_{im} C_{im}, r \geq r_w, \tag{1b}$$

$$\theta_{um} R_{um} \frac{\partial C_{um}}{\partial t} = \theta_{um} D_u \frac{\partial^2 C_{um}}{\partial z^2} \mp \theta_{um} v_{um} \frac{\partial C_{um}}{\partial z} - \omega_u(C_{um} - C_{uim}) - \theta_{um} \mu_{um} C_{um}, z \geq B, \tag{2a}$$

$$\theta_{uim} R_{uim} \frac{\partial C_{uim}}{\partial t} = \omega_u(C_{um} - C_{uim}) - \theta_{uim} \mu_{uim} C_{uim}, z \geq B, \tag{2b}$$

$$\theta_{lm} R_{lm} \frac{\partial C_{lm}}{\partial t} = \theta_{lm} D_l \frac{\partial^2 C_{lm}}{\partial z^2} \pm \theta_{lm} v_{lm} \frac{\partial C_{lm}}{\partial z} - \omega_l(C_{lm} - C_{lim}) - \theta_{lm} \mu_{lm} C_{lm}, z \leq -B, \tag{3a}$$

$$\theta_{lim} R_{lim} \frac{\partial C_{lim}}{\partial t} = \omega_l(C_{lm} - C_{lim}) - \theta_{lim} \mu_{lim} C_{lim}, z \leq -B, \tag{3b}$$

where subscript ''$u$'' refers to parameters in the upper aquitard; subscript "$l$" refers to parameters in the lower aquitard; subscript ''$m$'' refers to parameters in the mobile domain; subscript "$im$" refers to parameters in the immobile domains; $C_m$ and $C_{im}$ are the concentrations [ML$^{-3}$] of the aquifer; $C_{um}$ and $C_{uim}$ are concentrations [ML$^{-3}$] of the upper aquitard; $C_{lm}$ and $C_{lim}$ are concentrations [ML$^{-3}$] of the lower aquitard; $t$ is the time [T]; $B$ is half of the aquifer thickness [L]; $r$ is the radial distance [L]; $z$ represents the vertical distance [L]; $r_w$ is the well radius [L]; $D_r$ is aquifer dispersion coefficient [L$^2$T$^{-1}$]; $D_u$

and $D_l$ are vertical dispersion coefficients [L$^2$T$^{-1}$] of the upper aquitard and lower aquitard, respectively; $v_a$ is average velocity [LT$^{-1}$] and $v_a = \frac{u_a}{\theta_m}$; $u_a$ is Darcian velocity [LT$^{-1}$]; $v_{um}$ and $v_{lm}$ are vertical velocities [LT$^{-1}$]; $\mu_m, \mu_{im}, \mu_{um}, \mu_{uim}$, $\mu_{lm}$ and $\mu_{lim}$ are reaction rates; $\theta_m$, $\theta_{im}$, $\theta_{um}$, $\theta_{uim}$, $\theta_{lm}$ and $\theta_{lim}$ are the porosities [dimensionless]; $R_m = 1 + \frac{\rho_b K_d}{\theta_m}$, $R_{im} = 1 + \frac{\rho_b K_d}{\theta_{im}}$, $R_{um} = 1 + \frac{\rho_b K_d}{\theta_{um}}$, $R_{uim} = 1 + \frac{\rho_b K_d}{\theta_{uim}}$, $R_{lm} = 1 + \frac{\rho_b K_d}{\theta_{lm}}$ and $R_{lim} = 1 + \frac{\rho_b K_d}{\theta_{lim}}$ are the retardation factors





[dimensionless]; $K_d$ is the equilibrium distribution coefficient [M$^{-1}$L$^3$]; $\rho_b$ is the bulk density [ML$^{-3}$]; $\omega_a$, $\omega_u$ and $\omega_l$ are the first-order mass transfer coefficients [T$^{-1}$].

The symbol of the advection term is positive in the extraction phase in above equations, while it is negative before that. The dispersions are assumed to be linearly changing with the flow velocity, and one has:

$$D_r = \alpha_r |v_r| + D_r^*, \tag{4a}$$

$$D_u = \alpha_u |v_u| + D_u^*, \tag{4b}$$

$$D_l = \alpha_l |v_l| + D_l^*, \tag{4c}$$

where $\alpha_r$, $\alpha_u$ and $\alpha_l$ are dispersivities [L]; $D_r^*$, $D_u^*$ and $D_l^*$ are the diffusion coefficients [L$^2$T$^{-1}$].

Initial conditions are:

$$C_m(r,t)|_{t=0} = C_{im}(r,t)|_{t=0} = C_{um}(r,z,t)|_{t=0} = C_{uim}(r,z,t)|_{t=0} = C_{lm}(r,z,t)|_{t=0} = C_{lim}(r,z,t)|_{t=0} = 0, r \geq r_w, \tag{5}$$

The boundary conditions at infinity are:

$$C_m(r,t)|_{r\to\infty} = C_{im}(r,t)|_{r\to\infty} = C_{um}(r,z,t)|_{z\to\infty} = C_{uim}(r,z,t)|_{z\to\infty} = C_{lm}(r,z,t)|_{z\to-\infty} = C_{lim}(r,z,t)|_{z\to-\infty},$$

$$= 0, r \geq r_w, \tag{6}$$

Due to the concentration continuity at the aquifer-aquitard interface, one has:

$$C_m(r,t) = C_{um}(r,z=B,t), \tag{7a}$$

$$C_m(r,t) = C_{lm}(r,z=-B,t). \tag{7b}$$

The flux concentration continuity (FCC) is applied on the surface of wellbore, and one has:

$$\left[v_a C_m(r,t) - \alpha_r |v_a| \frac{\partial C_m(r,t)}{\partial r}\right]\Big|_{r=r_w} = \left[v_a C_{inj,m}(t)\right]\Big|_{r=r_w}, 0 < t \leq t_{inj}, \tag{8}$$

$$\left[v_a C_m(r,t) - \alpha_r |v_a| \frac{\partial C_m(r,t)}{\partial r}\right]\Big|_{r=r_w} = \left[v_a C_{cha,m}(t)\right]\Big|_{r=r_w}, t_{inj} < t \leq t_{cha}, \tag{9}$$

$$[C_m(r,t)]|_{r=r_w} = C_{res,m}(r_w,t), t_{cha} < t \leq t_{res}, \tag{10}$$

$$\left[v_a C_m(r,t) - \alpha_r |v_a| \frac{\partial C_m(r,t)}{\partial r}\right]\Big|_{r=r_w} = \left[v_a C_{ext,m}(t)\right]\Big|_{r=r_w}, t_{res} < t \leq t_{ext}, \tag{11}$$

where $t_{inj}$, $t_{cha}$, $t_{res}$ and $t_{ext}$ are the end moments [T] of the injection phase, the chaser phase, the rest phase and the extraction phase, respectively; $C_{inj,m}(t)$, $C_{cha,m}(t)$, $C_{res,m}(t)$ and $C_{ext,m}(t)$ represent the wellbore concentrations [ML$^{-3}$] of tracer in the injection phase, the chaser phase, the rest phase and the extraction phase, respectively.

The variation of the concentration with mixing effect in the injection phase could be described by (Wang et al., 2018):

$$V_{w,inj} \frac{dC_{inj,m}}{dt} = -\xi v_a(r_w)\left[C_{inj,m}(t) - C_0\right], 0 < t \leq t_{inj}, \tag{12a}$$

$$C_{inj,m}(t)\big|_{t=0} = 0, 0 < t \leq t_{inj}, \tag{12b}$$

$$V_{w,inj} = \pi r_w^2 h_{w,inj}, \tag{12c}$$

$$\xi = 2\pi r_w \theta 2B, \tag{12d}$$

where $h_{w,inj}$ is the wellbore water depth [L] in the injection phase.





As for the chaser phase, the models describing the concentration variation in the wellbore could be obtained using mass
balance principle:

$$V_{w,cha} \frac{dC_{cha,m}}{dt} = -\xi v_a(r_w)[C_{cha,m}(t)], t_{inj} < t \le t_{cha}, \tag{13a}$$

$$C_{cha,m}(t)|_{t=t_{inj}} = C_{inj,m}(t)|_{t=t_{inj}}, t_{inj} < t \le t_{cha}, \tag{13b}$$

$$V_{w,cha} = \pi r_w^2 h_{w,cha}, \tag{13c}$$

where $h_{w,cha}$ is the wellbore water depth [L] in the chaser phase.

In the extraction phase, the boundary condition is (Wang et al., 2018):

$$V_{w,ext} \frac{dC_{ext,m}}{dt}\bigg|_{r=r_w} = -\xi \alpha_r v_a(r_w) \frac{dC_{ext,m}}{dt}\bigg|_{r=r_w}, t_{res} < t \le t_{ext}, \tag{14a}$$

$$C_{ext,m}(t)|_{t=t_{res}} = C_{res,m}(t)|_{t=t_{res}}, t_{res} < t \le t_{ext}, \tag{14b}$$

$$V_{w,ext} = \pi r_w^2 h_{w,ext}, \tag{14c}$$

where $h_{w,ext}$ is the wellbore water depth [L] in the extraction phase.

## 2.2 Flow field model

The flow problem must be solved first before investigating the transport problem of the SWIW test. The velocity involved in
the advection and dispersion terms of the governing equations (1a) and (1b) is:

$$v_a(r_w) = \frac{Q}{4\pi r_w B \theta_m}, r \ge r_w, \tag{15}$$

where $Q$ is the pumping rate [L$^3$T$^{-1}$], and it is negative for injection and positive for pumping. The use of Eq. (15) implies
that quasi-steady state flow can be established very quickly near the injection/pumping well, thus the flow velocity becomes
independent of time. This approximation is generally acceptable given the very limited spatial range of influence of most
SWIW tests. For instance, if the characteristic length of SWIW test is $l$ and the aquifer hydraulic diffusivity is $D=K_a/S_a$,
where $K_a$ are $S_a$ are the radial hydraulic conductivity and specific storage, then the typical characteristic time of unsteady-
state flow is around $t_c \approx \frac{l^2}{2D}$. For instance, for a typical $l$=10 m, $K_a$=10 m/day and $S_a$=10$^{-5}$ (m$^{-1}$) (which are representative of
an aquifer consisting of medium sands), the value of $t_c$ is found to be $5.0 \times 10^{-5}$ day.

The water levels in the wellbore in Eqs. (12) - (14) could be calculated by the models of Moench (1985):

$$h_w = \lim_{t \to \infty} \{\mathcal{L}^{-1}[\bar{h}_w(p)]\}, \tag{16}$$

where $p$ is Laplace transform variable; $\mathcal{L}^{-1}$ represents the inverse Laplace transform; the over bar represents the Laplace-
domain variable, and

$$\bar{h}_w(p) = h_0 - \frac{Q}{8\pi KB} \frac{2[K_0(x) + xS_w K_1(x)]}{p\{pW_D[K_0(x) + xS_w K_1(x)] + xK_1(x)\}}, \tag{17}$$

$$W_D = \frac{1}{4BS_a}, \tag{18}$$

$$x = \frac{(p+\bar{q})}{2}, \tag{19}$$





$$\bar{q} = (\gamma')^2 m' \coth(m') + (\gamma'')^2 m'' \coth(m''), \tag{20}$$

$$m' = \frac{\left(\frac{S_u B_u p}{S_a B}\right)^{1/2}}{\gamma'}, \tag{21}$$

$$m'' = \frac{\left(\frac{S_l B_l p}{S_a B}\right)^{1/2}}{\gamma''}, \tag{22}$$

$$\gamma' = r_w \left(\frac{K_u}{2K_a B B_u}\right)^{1/2}, \tag{23}$$

$$\gamma'' = r_w \left(\frac{K_l}{2K_a B B_l}\right)^{1/2}, \tag{24}$$

where $K_u$ and $K_l$ are hydraulic conductivities [LT$^{-1}$]; $S_u$ and $S_l$ are specific storages [L$^{-1}$]; $S_w$ is the wellbore skin factor [dimensionless]; $B_u$ and $B_l$ are thicknesses [L]; $K_0(\cdot)$ and $K_1(\cdot)$ are the modified Bessel functions.

## 3 New solution of reactive transport in the SWIW test

In this study, the Laplace transform and Green's function methods will be employed to derive the analytical solution of the new SWIW test models described in Section 2. The dimensionless parameters are defined as follows: $C_{mD} = \frac{C_m}{C_0}$, $C_{imD} = \frac{C_{im}}{C_0}$, $C_{inj,mD} = \frac{C_{inj,m}}{C_0}$, $C_{inj,imD} = \frac{C_{inj,im}}{C_0}$, $C_{cha,mD} = \frac{C_{cha,m}}{C_0}$, $C_{cha,imD} = \frac{C_{cha,im}}{C_0}$, $C_{res,mD} = \frac{C_{res,m}}{C_0}$, $C_{res,imD} = \frac{C_{res,im}}{C_0}$, $C_{ext,mD} = \frac{C_{ext,m}}{C_0}$, $C_{ext,imD} = \frac{C_{ext,im}}{C_0}$, $C_{umD} = \frac{C_{um}}{C_0}$, $C_{uimD} = \frac{C_{uim}}{C_0}$, $C_{lmD} = \frac{C_{lm}}{C_0}$, $C_{limD} = \frac{C_{lim}}{C_0}$, $t_D = \frac{|A|}{\alpha_r^2 R_m} t$, $r_D = \frac{r}{\alpha_r}$, $r_{wD} = \frac{r_w}{\alpha_r}$, $z_D = \frac{z}{B}$, $\mu_{mD} = \frac{\alpha_r^2 \mu_m}{A}$, $\mu_{imD} = \frac{\alpha_r^2 R_m \mu_{im}}{R_{im} A}$, $\mu_{umD} = \frac{\alpha_r^2 \mu_{um}}{A}$, $\mu_{uimD} = \frac{\alpha_r^2 R_m \mu_{uim}}{R_{im} A}$, $\mu_{lmD} = \frac{\alpha_r^2 \mu_{lm}}{A}$, $\mu_{limD} = \frac{\alpha_r^2 R_m \mu_{lim}}{R_{im} A}$ and $A = \frac{Q}{4\pi B \theta_m}$. The detailed derivation of the new solution is listed in Section S1 of ***Supplementary Materials***.

### 3.1 Solutions in Laplace domain

As for the injection phase of the SWIW test, the solutions in Laplace domain are:

$$\bar{C}_{mD}(r_D, s) = \phi_1 \exp\left(\frac{y_{inj}}{2}\right) A_i\left(E^{1/3} y_{inj}\right), r_D \geq r_{wD}, \tag{25a}$$

$$\bar{C}_{imD} = \frac{\varepsilon_{im}}{(s + \mu_{imD} + \varepsilon_{im})} \bar{C}_{mD}, r_D \geq r_{wD}, \tag{25b}$$

$$\bar{C}_{umD} = \bar{C}_{mD} exp(a_2 z_D - a_2), z_D \geq 1, \tag{25c}$$

$$\bar{C}_{uimD} = \frac{\varepsilon_{uim}}{s + \varepsilon_{uim} + \mu_{uimD}} \bar{C}_{umD}, z_D \geq 1, \tag{25d}$$

$$\bar{C}_{lmD} = \bar{C}_{mD} exp(b_1 z_D + b_1), z_D \leq -1, \tag{25e}$$

$$\bar{C}_{limD} = \frac{\varepsilon_{lim}}{s + \varepsilon_{lim} + \mu_{limD}} \bar{C}_{lmD}, z_D \leq -1, \tag{25f}$$





where $s$ represents the Laplace transform parameter for $t_D$ (which is proportional to $p$); $A_i(\cdot)$ is the Airy function $A'_i(\cdot)$ is the derivative of the Airy function; the expressions for $a_2$, $b_1$, $E$, $y_{inj}$, $y_{inj,w}$, $\varepsilon_m$, $\varepsilon_{im}$, $\varepsilon_{um}$, $\varepsilon_{uim}$, $\varepsilon_{lm}$, $\varepsilon_{lim}$, $\beta_{inj}$ and $\phi_1$ are listed in Table 1.

In the chaser phase, the solutions of the SWIW test in Laplace domain are:

$$\bar{C}_{mD} = \Psi(r_D) + \delta_1 + \delta_2 r_D, r_D \geq r_{wD}, \tag{26a}$$

$$\bar{C}_{imD} = \frac{\varepsilon_{im}}{(s+\mu_{imD}+\varepsilon_{im})}\bar{C}_{mD} + \frac{C_{imD}(r_D,t_{inj,D})}{(s+\mu_{imD}+\varepsilon_{im})}, r_D \geq r_{wD}, \tag{26b}$$

$$\Psi(r_D, E_a; \eta) = \int_{r_{wD}}^{\infty} g(r_D, E_a; \eta)\, \varphi(\eta)d\eta, r_D \geq r_{wD}, \tag{26c}$$

$$\bar{C}_{umD} = \int_1^{\infty} g_u(z_D, E_u; \eta_u) f_u(\eta_u)d\eta_u + \frac{z_D-z_{eD}}{1-z_{eD}}\bar{C}_{mD}(r_D, s), z_D \geq 1, \tag{26d}$$

$$\bar{C}_{uimD} = \frac{\varepsilon_{uim}}{s+\varepsilon_{uim}+\mu_{uimD}}\bar{C}_{umD} + \frac{C_{uimD}(r_D,z_D,t_{inj,D})}{s+\varepsilon_{uim}+\mu_{uimD}}, z_D \geq 1, \tag{26e}$$

$$\bar{C}_{lmD} = \int_{-1}^{-\infty} g_l(z_D, E_l; \eta_l)f_l(\eta_l)d\eta_l + \frac{z_{eD}+z_D}{z_{eD}-1}\bar{C}_{mD}(r_D, z_D, s), z_D \leq -1, \tag{26f}$$

$$\bar{C}_{limD} = \frac{\varepsilon_{lim}}{s+\varepsilon_{lim}+\mu_{limD}}\bar{C}_{lmD} + \frac{C_{limD}(r_D,z_D,t_{inj,D})}{s+\varepsilon_{lim}+\mu_{limD}}, z_D \leq -1, \tag{26g}$$

where $\eta$ varies between $r_{wD}$ and $\infty$, e.g. $r_{wD} \leq \eta \leq \infty$; $\eta_u$ varies between 1 and $\infty$; $\eta_l$ varies between $-1$ and $-\infty$; $C_{mD}(r_D, t_{inj,D})$ and $C_{imD}(r_D, t_{inj,D})$ are the concentrations [ML⁻³] of the aquifer at the end of injection stage, which could be calculated by Eq. (25a) and Eq. (25b) after applying the inverse Laplace transform, $C_{umD}(r_D, z_D, t_{inj,D})$ and $C_{uimD}(r_D, z_D, t_{cha,D})$ represent the concentrations [ML⁻³] of the upper aquitard at the end of the injection phase, which could

be calculated by Eq. (25c) and Eq. (25d) after applying the inverse Laplace transform, $C_{lmD}(r_D, z_D, t_{inj,D})$ and $C_{limD}(r_D, z_D, t_{inj,D})$ are the concentrations [ML⁻³] of the lower aquitard at the end of the injection phase, which could be calculated by Eq. (25e) and Eq. (25f) after applying the inverse Laplace transform, $g(r_D, E_a; \eta)$, $g_u(z_D, E_u; \eta_u)$ and $g_l(z_D, E_l; \eta_l)$ are the Green's functions; the expressions for $g(r_D, E_a; \eta)$, $g_u(z_D, E_u; \eta_u)$, $g_l(z_D, E_l; \eta_l)$, $\delta_1$, $\delta_2$, $s_1$, $s_2$, $E_a$, $E_u$, $E_l$, $y_{cha}$, $y_{cha,w}$, $F$, $\varphi(r_D)$, $f_u(\eta_u)$, $X$, $M_1$, $M_2$, $M_3$, $M_4$, $N_1$, $N_2$, $N_3$, $N_4$, $\mathcal{T}_1$, $\mathcal{T}_2$, $\mathcal{T}_3$, $\mathcal{T}_4$ and $\beta_{cha,D}$ are listed in Table 2.

For the rest phase, the solutions of the SWIW test in Laplace domain are:

$$\bar{C}_{mD} = \frac{C_{mD}(r_D,t_{cha,D}) + \frac{\varepsilon_m C_{imD}(r_D,t_{cha,D})}{(s+\mu_{imD}+\varepsilon_{im})}}{\left(s+\varepsilon_m+\mu_{mD}-\frac{\varepsilon_m\varepsilon_{im}}{s+\mu_{imD}+\varepsilon_{im}}\right)}, r_D \geq r_{wD}, \tag{27a}$$

$$\bar{C}_{imD} = \frac{C_{imD}(r_D,t_{cha,D})}{(s+\mu_{imD}+\varepsilon_{im})} + \frac{\varepsilon_{im}\bar{C}_{mD}}{(s+\mu_{imD}+\varepsilon_{im})}, r_D \geq r_{wD}, \tag{27b}$$

$$\bar{C}_{umD} = \frac{C_{umD}(r_D,z_D,t_{cha,D}) + \frac{\varepsilon_{um} C_{uimD}(r_D,z_D,t_{cha,D})}{s+\varepsilon_{uim}+\mu_{uimD}}}{\left(s+\varepsilon_{um}+\mu_{umD}-\frac{\varepsilon_{um}\varepsilon_{uim}}{s+\varepsilon_{uim}+\mu_{uimD}}\right)}, z_D \geq 1, \tag{27c}$$

$$\bar{C}_{uimD} = \frac{\varepsilon_{uim}}{s+\varepsilon_{uim}+\mu_{umD}}\bar{C}_{umD} + \frac{C_{uimD}(r_D,z_D,t_{cha,D})}{s+\varepsilon_{uim}+\mu_{umD}}, z_D \geq 1, \tag{27d}$$



$\bar{C}_{lmD} = \dfrac{C_{lmD}(r_D, z_D, t_{cha,D}) + \frac{\varepsilon_{lm} C_{limD}(r_D, z_D, t_{cha,D})}{s + \varepsilon_{lim} + \mu_{limD}}}{\left(s + \varepsilon_{lm} + \mu_{lmD} - \frac{\varepsilon_{lm}\varepsilon_{lim}}{s + \varepsilon_{lim} + \mu_{limD}}\right)}, z_D \leq -1,$ (27e)

$\bar{C}_{limD} = \dfrac{\varepsilon_{lim}}{s + \varepsilon_{lim} + \mu_{lmD}} \bar{C}_{lmD} + \dfrac{C_{limD}(r_D, z_D, t_{cha,D})}{s + \varepsilon_{lim} + \mu_{lmD}}, z_D \leq -1,$ (27f)

where $C_{mD}(r_D, t_{cha,D})$ and $C_{imD}(r_D, t_{cha,D})$ are the concentrations [ML$^{-3}$] of the aquifer at the end of the chaser phase, which could be calculated by Eq. (26a) and Eq. (26b) after applying the inverse Laplace transform, $C_{umD}(r_D, z_D, t_{cha,D})$ and $C_{uimD}(r_D, z_D, t_{cha,D})$ are the concentrations [ML$^{-3}$] of the upper aquitard at the end of the chaser phase, which could be

computed by Eq. (26d) and Eq. (26e) after applying the inverse Laplace transform, $C_{lmD}(r_D, z_D, t_{cha,D})$ and $C_{limD}(r_D, z_D, t_{cha,D})$ are the concentrations [ML$^{-3}$] of the lower aquitard at the end of the chaser phase, which could be calculated by Eq. (26f) and Eq. (26g) after applying the inverse Laplace transform.

As for the extraction phase of the SWIW test, the solutions in Laplace domain are:

$\bar{C}_{mD}(r_D, s) = exp(-r_D/2)[U(r_D, \zeta; \varepsilon) + \sigma_1 + \sigma_2 r_D], r_D \geq r_{wD},$ (28a)

$\bar{C}_{imD} = \dfrac{\varepsilon_{im}}{(s + \mu_{imD} + \varepsilon_{im})} \bar{C}_{mD} + \dfrac{C_{imD}(r_D, res)}{s + \mu_{imD} + \varepsilon_{im}}, r_D \geq r_{wD},$ (28b)

$U(r_D, \zeta; \varepsilon) = \int_{r_{wD}}^{\infty} g(r_D, \zeta; \varepsilon) f(\varepsilon) d\varepsilon,$ (28c)

$\bar{C}_{umD} = \int_1^{\infty} g_u(z_D, E_u; \mathcal{b}_u) f_u(\mathcal{b}_u) d\mathcal{b}_u + \dfrac{z_D - z_{eD}}{1 - z_{eD}} \bar{C}_{mD}(r_D, s), z_D \geq 1,$ (28d)

$\bar{C}_{uimD} = \dfrac{\varepsilon_{uim} \bar{C}_{umD}}{s + \varepsilon_{uim} + \mu_{uimD}} + \dfrac{C_{uimD}(r_D, z_D, t_{res,D})}{s + \varepsilon_{uim} + \mu_{uimD}}, z_D \geq 1,$ (28e)

$\bar{C}_{lmD} = \int_{-1}^{-\infty} g_l(z_D, E_l; \mathcal{b}_l) f_l(\mathcal{b}_l) d\mathcal{b}_l + \dfrac{z_D + z_{eD}}{z_{eD} - 1} \bar{C}_{mD}(r_D, s), z_D \leq -1,$ (28f)

$\bar{C}_{limD} = \dfrac{\varepsilon_{lim} \bar{C}_{lmD}}{s + \varepsilon_{lim} + \mu_{limD}} + \dfrac{C_{limD}(r_D, z_D, t_{res,D})}{s + \varepsilon_{lim} + \mu_{limD}}, z_D \leq -1,$ (28g)

where $C_{mD}(r_D, t_{res,D})$ and $C_{imD}(r_D, t_{res,D})$ are the concentrations [ML$^{-3}$] of the aquifer at the end of the rest phase, which could be calculated by Eq. (27a) and Eq. (27b) after applying the inverse Laplace transform, $C_{umD}(r_D, z_D, t_{res,D})$ and $C_{uimD}(r_D, z_D, t_{res,D})$ are the concentrations [ML$^{-3}$] of the upper aquitard at the end of the rest phase, which could be calculated by Eq. (27c) and Eq. (27d) after applying the inverse Laplace transform, $C_{lmD}(r_D, z_D, t_{res,D})$ and

$C_{limD}(r_D, z_D, t_{res,D})$ are the concentrations [ML$^{-3}$] of the lower aquitard at the end of the rest phase, which could be calculated by Eq. (27e) and Eq. (27f) after applying the inverse Laplace transform; $\mathcal{b}_u$ varies between 1 and $\infty$; $\mathcal{b}_l$ varies between $-1$ and $-\infty$; $\varepsilon$ varies between $r_{wD}$ and $\infty$ (e.g. $r_{wD} \leq \varepsilon \leq \infty$); $g(r_D, \zeta; \varepsilon)$, $g_u(z_D, E_u; \mathcal{b}_u)$ and $g_l(z_D, E_l; \mathcal{b}_l)$ are the Green's functions; the expressions for $g(r_D, \zeta; \varepsilon)$, $g_u(z_D, E_u; \mathcal{b}_u)$, $g_l(z_D, E_l; \mathcal{b}_l)$, $\sigma_1$, $\sigma_2$, $\Lambda$, $\zeta$, $f(\varepsilon)$, $f_u(\mathcal{b}_u)$, $f_l(\mathcal{b}_l)$, $H_1 \sim H_4$, $I_1 \sim I_4$, $m_1 \sim m_2$, $n_1 \sim n_2$, $P_1 \sim P_4$, W, $y_{ext}$, $y_{ext,w}$ and $\beta_{ext,D}$ are listed in Table 3.

The new SWIW model is a generalization of many previous models. For instance, when both mixing effect and aquitard effect are excluded, the new model of this study becomes Chen et al. (2017). When excluding both effects of aquitard and





well mixing, and reducing the four-phase SWIW test into a two-phase SWIW test, the new model this study becomes Gelhar and Collins (1971).

**3.2 Solutions from Laplace domain to real-time domain**

Because the analytical solutions in Laplace domain are too complex, it seems impossible to transform it into the real time domain analytically. Alternatively, a numerical method will be introduced for the invers Laplace transform. Currently, several methods are available, like the Stehfest model, Zakian model, Fourier series model, de Hoog model, and Schapery model (Wang and Zhan, 2015). Here, the de Hoog method will be applied to conduct the inverse Laplace transform, since it performed well for radial-dispersion problems (Wang et al., 2018;Wang and Zhan, 2013).

**3.3 Assumptions included in the new SWIW test model**

Although the new SWIW test model is a generalization of many previous studies, three assumptions still remain. First, the flow is in the steady state, e.g. Eq. (15). Second, the groundwater flow is horizontal in the aquifer, and is vertical in the aquitard. This treatment relies on the basis that the permeability of the aquitard is smaller than the permeability of the aquifer (Moench, 1985). Third, the model is simplified for the solute transport. For example, only vertical dispersion and advection

effects are considered in the aquitard, and only radial dispersion and advection effects are considered in the aquifer. The validation of these assumptions will be discussed in the Section 4.2.

**4 Verification of the new model**

**4.1 Test of the new solution with previous solutions**

To test the new solutions, the model of Chen et al. (2017) serves as a benchmark, who ignored the aquitard effect and

wellbore mixing effect in the SWIW test. Figure 1 shows the comparison of BTCs between them, and the parameters used in such a comparison are: $R_m = R_{im} = R_{um} = R_{uim} = R_{lm} = R_{lim} = 1$, $\theta_{um} = \theta_{lm} = 0.1$, $\alpha_r = \alpha_u = \alpha_l = 0.1$m, $\mu_m = \mu_{im} = \mu_{um} = \mu_{uim} = \mu_{lm} = \mu_{lim} = 10^{-6}$d$^{-1}$, $r_w = 0.2$m, $Q_{inj} = 2.5$ m$^3$/d, $Q_{cha} = 2.5$ m$^3$/d, $Q_{res} = 0$ m$^3$/d, $Q_{ext} = -2.5$ m$^3$/d, $t_{inj} = 100$day, $t_{cha} = 50$day, $t_{res} = 40$day, $B = 5$ m, $\theta_m = 0.3$, $\theta_{im} = 0.15$, $\theta_{uim} = \theta_{lim} = 0.1$, and $\omega = 0$ d$^{-1}$. The parameters of "$h_{w,inj} = h_{w,cha} = h_{w,res} = h_{w,ext} = 0$" represent $V_{w,inj} = 0$, $V_{w,cha} = 0$ and $V_{w,ext} = 0$, and imply that the mixing effect in the wellbore is

neglected. The values of $v_{um} = v_{lm} = 0$ m/d mean that aquitards are neglected. As shown in Figure 1, both solutions agree well for the mobile and immobile domains.

**4.2 Test of assumptions involved in the analytical solution**

To test the three assumptions outlined in Section 3.3, a numerical model will be established, where general three-dimensional transient flow and solute transport are considered in both aquifer and aquitards. A finite-element method with





the help of COMSOL Multiphysics will be used to solve the three-dimensional model. The grid system is shown in Section

S2 of ***Supplementary Materials***.

In this study, four sets of aquitard hydraulic conductivities are employed, such as $K_u = K_l = 0.1 K_a$, $K_u = K_l = 0.02 K_a$, $K_u = K_l = 0.01 K_a$, and $K_u = K_l = 0.001 K_a$. A point to note is that the extreme case of $K_u = K_l = 0.1 K_a$ used here is only for the purpose of examining the robustness of comparison, while the real values of $K_u$ and $K_l$ are usually much lower than $0.1 K_a$. In

another word, the rest three cases mentioned above are more likely to occur in real applications.

The initial drawdown and the initial concentration are 0 for aquifer and aquitards. The hydraulic parameters are: $K_a = 0.1$ m/day, $S_a = S_u = S_l = 10^{-4}$ m$^{-1}$, and the other parameters are $R_m = R_{im} = R_{um} = R_{uim} = R_{lm} = R_{lim} = 1$, $\theta_{um} = \theta_{lm} = 0.1$, $\alpha_r = 2.5$m, $\alpha_u = \alpha_l = 0.5$m, $\mu_m = \mu_{im} = \mu_{um} = \mu_{uim} = \mu_{lm} = \mu_{lim} = 10^{-7}$s$^{-1}$, $r_w = 0.5$m, $Q_{inj} = Q_{cha} = 50$ m$^3$/d, $Q_{res} = 0$ m$^3$/d, $Q_{ext} = -50$ m$^3$/d, $t_{inj} = 250$day, $t_{cha} = 50$day, $t_{res} = 50$day, $B = 10$m, $\theta_m = 0.3$, $\theta_{im} = 0.0$, and $\omega = 0$ d$^{-1}$. The comparison of

concentration between the analytical and numerical solutions is shown in Figs. 2 and 3.

As the first assumption in Section 3.3 has been elaborated in Section 2.2, the following discussion will only focus on the second and third assumptions. Figs. 2a, 2b and 2c represent the snapshots of concentration distributions in the aquifer along the $r$-axis at different times. One may conclude that the curves with smaller $K_u$ and $K_l$ values are closer to the analytical solution. This is because aquitards with smaller $K_u$ and $K_l$ (when $K_a$ remains constant) could make flow closer to the

horizontal direction (or parallel with the aquitard-aquifer interface) in the aquifer and closer to the vertical direction (or perpendicular with the aquitard-aquifer interface) in the aquitard, according to the law of refraction (Fetter, 2018). In another word, when the values of $K_u/K_a$ and $K_l/K_a$ are 0, the flow direction becomes horizontal in the aquifer and vertical in the aquitard, and then the numerical model reduces to the analytical model. Therefore, from this figure, one may conclude that the above-mentioned second assumption in Section 3.3 works well in the aquifer when $K_u/K_a$ and $K_l/K_a$ are samller then

300 0.01.

Fig. 3 shows the comparison of the analytical and numerical solutions for aquitards. Figs. 3 (a1) - (c1) represent the snapshots of concentration distributions obtained from analytical solutions of this study at different times, and Figs. 3 (a2) - (c2) represent the snapshots of concentration distributions obtained from the numerical solutions at the same time. One may find that the contour maps obtained from both solutions are almost the same in the aquifer, but very different in the aquitards.

Therefore, the above-mentioned third assumption in Section 3.3 is generally unacceptable in describing solute transport in the aquitard in the SWIW test, but works well when the aquifer is of the primary concern.

## 5 Discussions

### 5.1 Model applications

As mentioned in Section 3.1, the new model is a generalization of many previous models, and the conceptual model is more

close to reality. However, there are many parameters involved in this new model that have to be determined first for applying





this model. For instance, the involved parameters for the aquitards include dispersivity ($\alpha_u$ and $\alpha_l$), first-order mass transfer coefficient ($\omega_u$ and $\omega_l$), retardation factor ($R_{um}$, $R_{uim}$, $R_{lm}$, and $R_{lim}$), porosity ($\theta_{um}$, $\theta_{uim}$, $\theta_{lm}$ and $\theta_{lim}$), reaction rate ($\mu_{um}$, $\mu_{uim}$, $\mu_{lm}$ and $\mu_{lim}$), and velocity ($v_{um}$ and $v_{lm}$). The involved parameters for the aquifer include $\alpha_r$, $\omega_a$, $R_m$, $R_{im}$, $\theta_m$, $\theta_{im}$, and $B$. Generally, these parameters could be measured directly. Otherwise, they could be obtained by fitting the experimental data using the forward model.

Parameter estimation is an inverse problem, and it is generally conducted by an optimization model, such as genetic algorithm, simulated annealing, and so on. Due to the ill-posedness of many inverse problems or insufficient observation data, the initial guess values of unknown parameters of interest are critical for finding the best values or real values of those parameters in the optimization model. Here, we recommend using values of parameters from literatures as the initial guesses for similar lithology. Table 4 lists some parameter values for sandy and clay aquifers in previous studies. When result is not sensitive to a particular parameter of concern, the value from previous publications for similar lithology and/or situations could be taken as estimated value of that parameter, if there is no direct measurement of that particular parameter of concern. To prioritize the sensitivity of parameters involved the new model, a sensitivity analysis is conducted in Section 5.2.

**5.2 Sensitivity analysis**

From the analytical solutions of Eqs. (26) - (28), one may find that BTCs are affected by several parameters, like $\alpha_u$, $v_{um}$, $\theta_{um}$, $\omega$, $\alpha_r$, $\theta_m$ and $V_w$. In this section, the sensitivity analysis is conducted, and the model is (Kabala, 2001;Yang and Yeh, 2009):

$$SC_{i,j} = I_j \frac{c_i(I_j + \Delta I_j) - c_i(I_j)}{\Delta I_j}, \tag{29}$$

where $\Delta I_j$ is a small increment of $I_j$; $SC_{i,j}$ is the sensitivity coefficient. $I_j$ could be any individual parameter of interest. A larger $|SC_{i,j}|$ value represents that the result is more sensitive to that particular parameter.

Fig. 4 represents the sensitivity coefficients of BTCs. One may find that the influence of $v_{um}$, $\theta_{um}$, and $\alpha_r$ on the results is more obvious than others. As the values of $v_{um}$ is genrously small, we mainly focus on the discussion of $\theta_{um}$, and $\alpha_r$ in the following sections.

**5.3 Effect of the aquitard**

As shown in Section 4.2, the new analytical solution is a good approximation for the numerical model in the aquifer when $K_u/K_a$ and $K_l/K_a$ are smaller then 0.01. In this section, we try to figure out how the aquitards will affect BTCs of the SWIW tests. Since the porosity is an important factor of concern, three sets of porosity values are used for the aquitards: $\theta_{um} = \theta_{lm} = 0$, 0.1, and 0.25. The other parameters are from the case in Figure 3.

Fig. 5 shows the difference between the models with and without aquitards for different flow velocities in the aquitard. The case of $\theta_{um} = \theta_{lm} = 0$ represents the model without the aquitard. The difference is not obvious at the beginning of the extraction phase, while such a difference is obvious at the late time. Meanwhile, the smaller aquitard porosity makes the





value of BTCs in the aquifer greater at a given time. When the aquitard is ignored, the values of BTCs are the greatest. Therefore, the aquitard effect on transport in the aquifer is quite obvious and should not be ignored in general.

**5.4 Effect of the aquifer radial dispersion**

Another important parameter is the radial dispersion in the aquifer. In this section, three sets of the radial dispersivity values will be used to analyze the influence: $\alpha_r$ =1.25m, 2.50m, and 5.00m.

Fig. 6 shows BTCs in the well face for different radial dispersivity values. Firstly, the difference is obvious among curves in all phases. Secondly, a larger $\alpha_r$ could decrease BTCs at a given time of the injection phase. This could be explained by the boundary condition of Eq. (8). The solute in the mobile domain of the aquifer is transported by both advection and dispersion,

thus a larger $\alpha_r$ could lower the values of $C_m$ in the well face. Thirdly, BTCs increase with increasing $\alpha_r$ values in the chaser and rest phases. Fourthly, the peak values of BTCs decrease with increasing $\alpha_r$ values.

**6 Data interpretation: Field SWIW test**

To test the performance of the new model, the field data reported in Chen et al. (2017) will be employed. Specifically, the experimental data of S1 conducted in the borehole TW3 will be analysed. The reason choosing this dataset is because this

borehole penetrated several layers, and it had been interpreted by Chen et al. (2017) before (using a model without considering the aquitard effect and the mixing effect). Detailed information of experimental data could be seen in the references of Assayag et al. (2009) and Yang et al. (2014).

Fig. 7 shows the fitness of the computed and observed BTCs. The parameters are: $R_m = R_{im} = R_{um} = R_{uim} = R_{lm} = R_{lim}$=1, $\theta_{um} = 0.05$, $\theta_{lm}$=0.0, $\alpha_r = 0.7134$m, $\alpha_u = 0.35$m, $\alpha_l$=0.0 m, $\mu_m = \mu_{im} = \mu_{um} = \mu_{uim} = \mu_{lm} = \mu_{lim}$=10⁻⁷s⁻¹, $r_w$

=0.1m, $Q_{inj} = Q_{cha}$ = 7.78L/min, $Q_{res}$ =0 L/min, $Q_{ext}$ =12 L/min, $t_{inj}$ =180min, $t_{cha}$ =26.74min, $t_{res}$ =10080min, $B$=8m, $\theta_m$=0.1, $\theta_{im}$=0.068, and $\omega$=0.001 d⁻¹. $h_{w,inj} = h_{w,cha}$ =32m, $h_{w,res}$=30m, $h_{w,ext}$=28m.

Apparently, the fitness by the new solution is better than the model of Chen et al. (2017). As for the error between the observed and computed BTCs, the new solution is also smaller than that of Chen et al. (2017) as well, where the error is defined as

$Error = \sum_{i=1}^{N}(C_{OBS} - C_{COM})^2,$ (30)

where $C_{OBS}$ and $C_{COM}$ are the observed and computed concentrations, respectively, and $N$ is the number of sampling points.

**7 Summary and conclusions**

The single-well injection-withdrawal (SWIW) test could be applied to estimate the dispersivity, porosity, chemical reaction rates of the *in situ* aquifers. However, previous studies mainly focused on an isolated aquifer, excluding all the possible

effect of aquitards bounding the aquifer. In another word, the adjacent layers are assumed to be non-permeable, which is not

exactly true in reality. In this study, a new analytical model is established and its associate solutions derived to inspect the effect of overlying and underlying aquitards. Meanwhile, four stages are considered in the new model, including the injection phase, the chaser phase, the rest phase and the extraction phase. The anomalous behaviors of reactive transport in the test were described by a mobile-immobile framework. The mixing effect was considered in the wellbore.

To derive the analytical solution of the new model, some assumptions are inevitable. For instance, only vertical advection and dispersion are considered in the aquitard and only horizontal advection and dispersion are considered in the aquifer, and the flow is quasi-steady state. Although these assumptions have been widely used to describe the radial dispersion in previous studies, the influences on reactive transport have not been discussed in a rigorous sense before. In this study, numerical modelling exercises will be introduced to test the above-mentioned assumptions of the new model. Based on this

study, the several conclusions could be obtained.

1. A new model of the SWIW test is a generalizing of many previous models by considering the aquitard effect, the

wellbore mixing effect, and the mass transfer rate in both aquifer and aquitards. The sub-model of the wellbore mixing effect

is developed.

2. Assumption of vertical advection and dispersion on the aquitard and horizontal advection and dispersion in the aquifer is

tested by specially designed finite-element numerical models using COMSOL and the result shows that this assumption is

acceptable when the aquifer is of primary concern, provided that the ratios of the aquitard/aquifer permeability are less than

0.01; while such an assumption is generally unacceptable when the aquitards are of concern, regardless of the ratios of the

aquitard/aquifer permeability.

3. The new model is most sensitive to the aquitard porosity and aquifer radial dispersivity after a comprehensive sensitivity

analysis. A larger aquifer radial dispersivity decreases BTCs in the injection stage, increases BTCs in the chaser and rest

stages. It decreases BTC peak values in the extraction stage.

4. The performance of the new model is better than previous models of excluding the aquitard effect and the wellbore mixing

effect in terms of best fitting exercises with field data reported in Chen et al. (2017).

**Acknowledgments**

This research was partially supported by Programs of Natural Science Foundation of China (No.41772252), and Innovative Research Groups of the National Nature Science Foundation of China (No. 41521001).





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

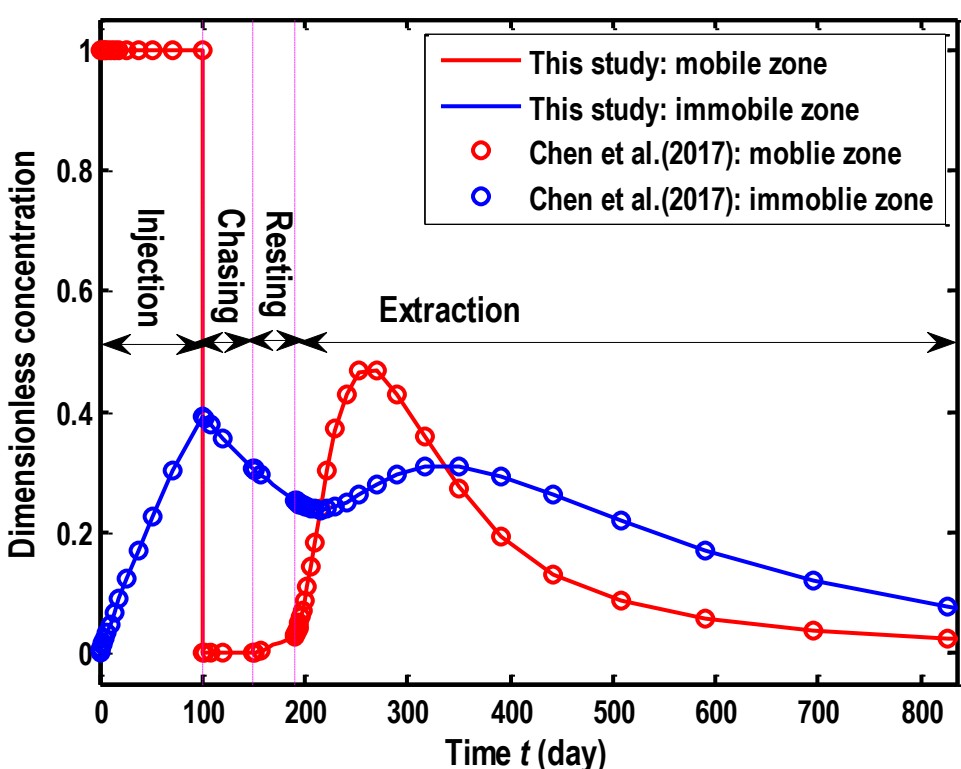

**Figure 1:  Comparison of BTCs at the well screen computed by the solution of this study and Chen et al. (2017).**





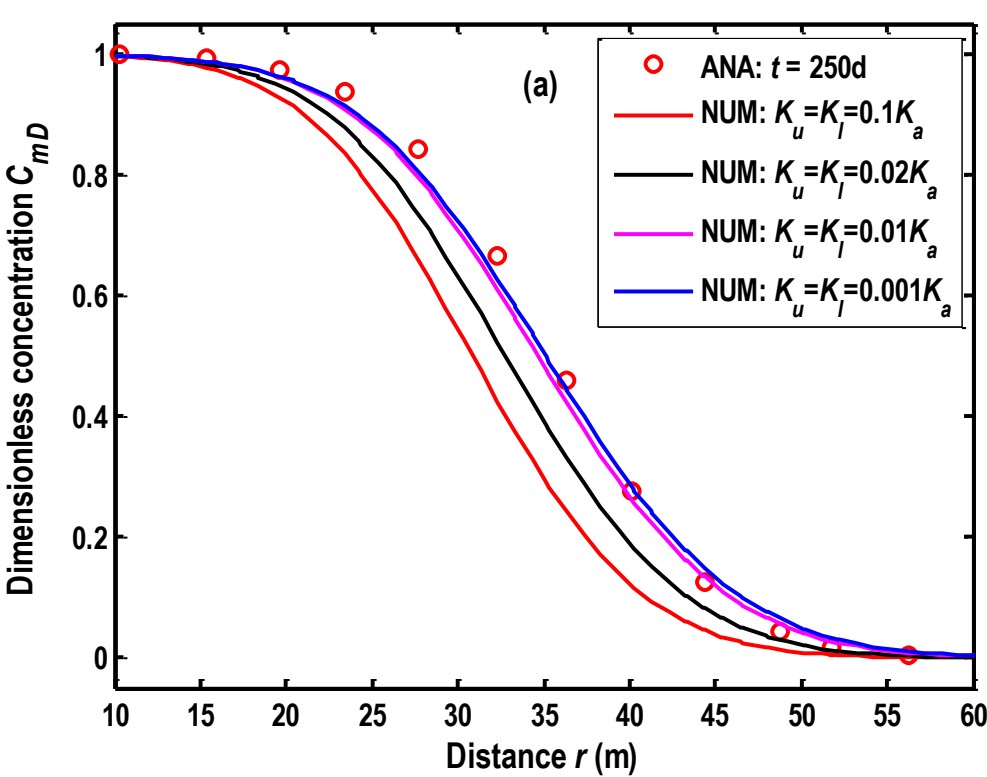


(a) At the end of the injection phase: $t = 250$ day





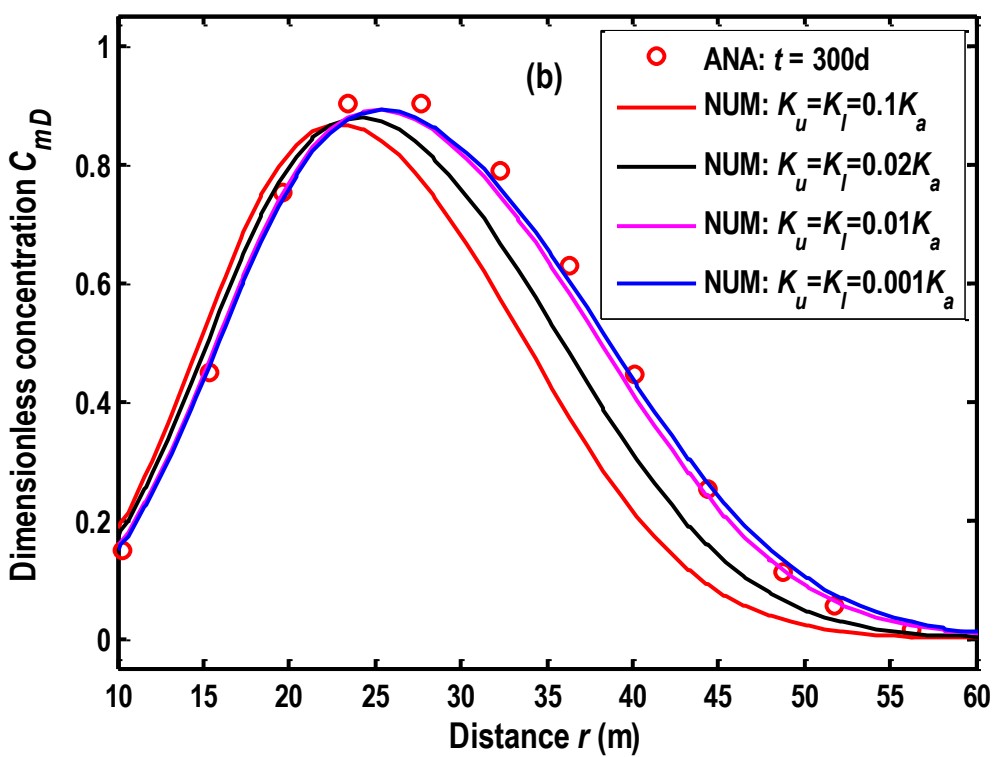

(b) At the end of the chasing phase: $t = 300$ day





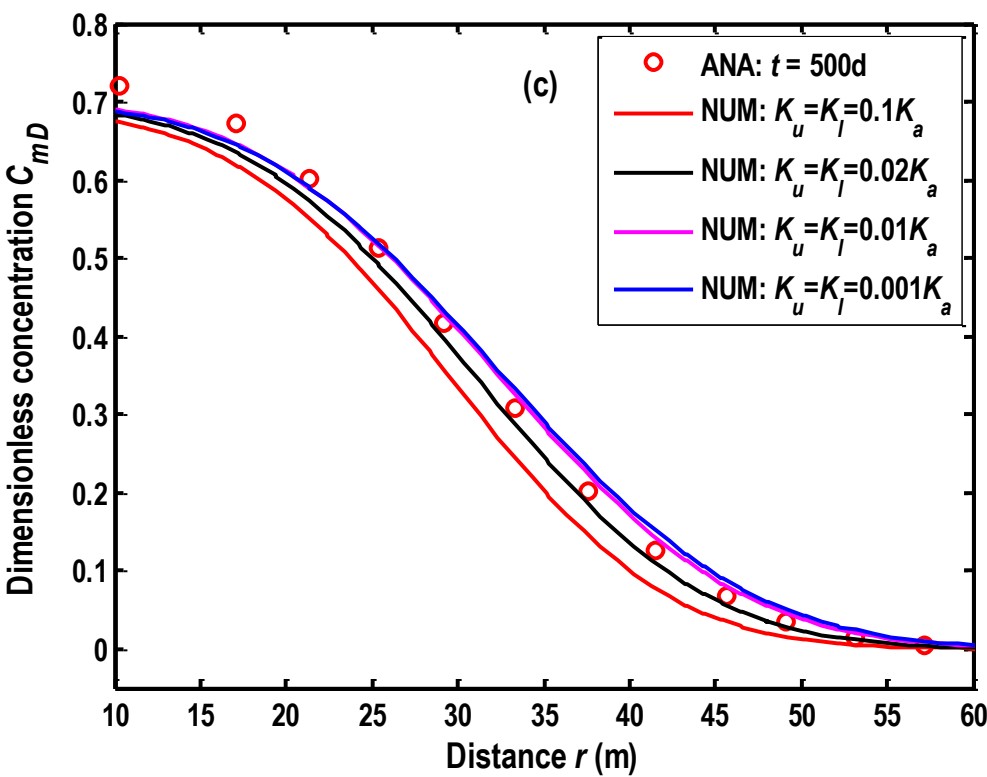

(c) In the extraction phase: $t = 500$ day

**Figure 2: Comparison of the concentration distribution between the analytical and numerical solutions along the $r$-axis at $z=0$m. "ANA" and "NUM" represent the analytical and numerical solutions, respectively.**



**Figure 3: The vertical profiles (the *r-z* profiles) of the concentrations. (a1) - (c1) represent the analycial solutions at *t*=250, 300 and 500 day, respectively. (a2) - (c2) represent the numerical solutions at *t*=250, 300 and 500 day, respectively.**



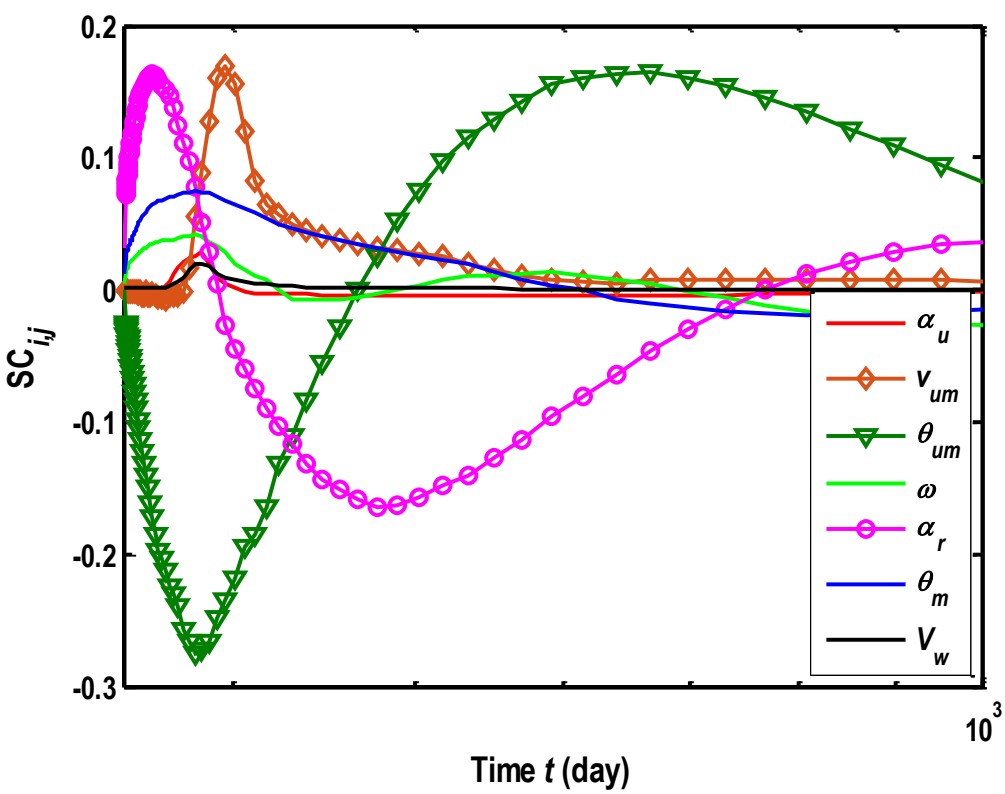

**Figure 4: SC$_{i,j}$ of the parameters $\alpha_u$, $v_{um}$, $\theta_{um}$, $\omega$, $\alpha_r$, $\theta_m$ and $V_w$ in the wellbore.**



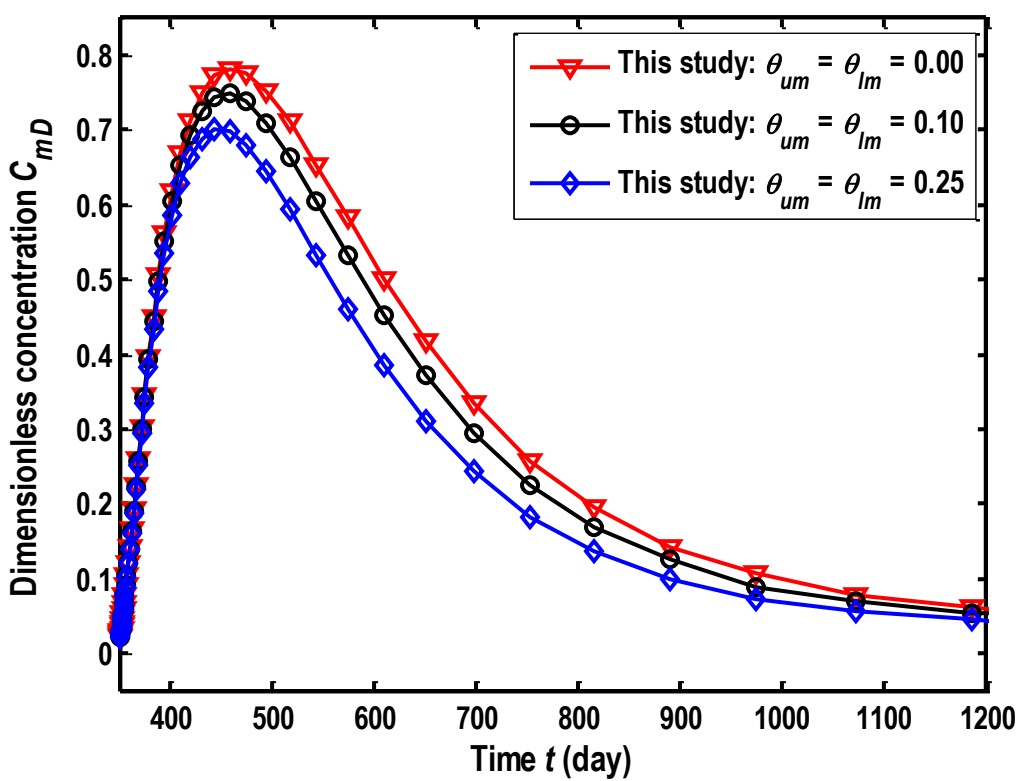

**Figure 5: Comparison of BTCs between the model with and without aquitards for different porosities.**

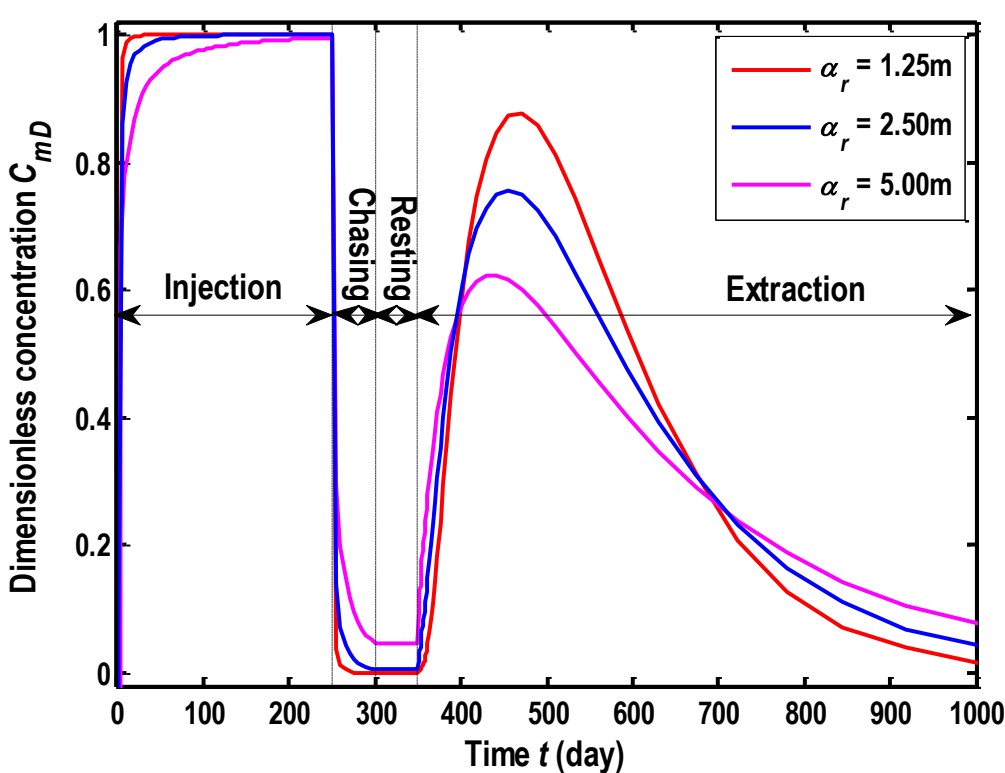


**Figure 6: BTCs in the wellbore for different $\alpha_r$.**

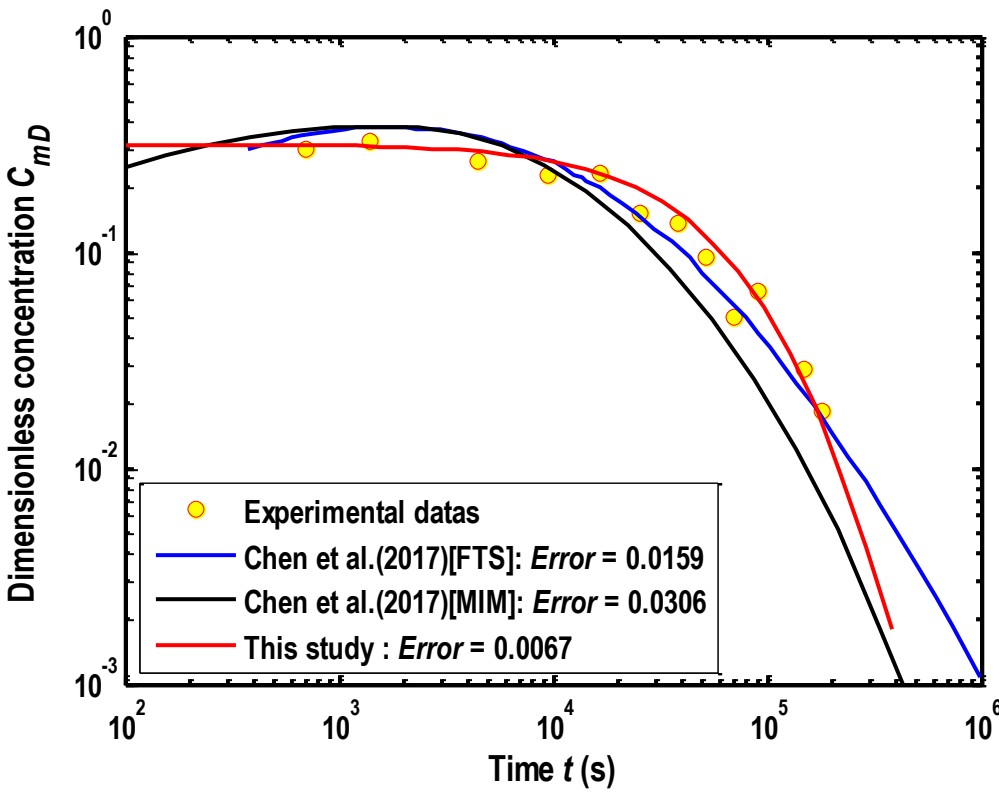

**Figure 7: Fitness of observed BTC by the previous model and new solutions of this study.**





**Table 1.:Expressions of the coefficients in the solutions expressed in Eqs.(25a) - (25f).**

| | |
|---|---|
| $a_2$ | $$\dfrac{\dfrac{R_m v_{um} \alpha_r^2}{ABR_{um}} - \sqrt{\left(\dfrac{R_m v_{um} \alpha_r^2}{ABR_{um}}\right)^2 + 4\dfrac{R_m \alpha_r^2 D_u}{AB^2 R_{um}}\left(s + \varepsilon_{um} + \mu_{umD} - \dfrac{\varepsilon_{um} \varepsilon_{uim}}{s + \mu_{uimD} + \varepsilon_{uim}}\right)}}{2\dfrac{R_m \alpha_r^2 D_u}{AB^2 R_{um}}}$$ |
| $b_1$ | $$\dfrac{-\dfrac{R_m v_{lm} \alpha_r^2}{ABR_{lm}} + \sqrt{\left(\dfrac{R_m v_{lm} \alpha_r^2}{ABR_{lm}}\right)^2 + 4\dfrac{R_m \alpha_r^2 D_l}{AB^2 R_{lm}}\left(s + \varepsilon_{lm} + \mu_{lmD} - \dfrac{\varepsilon_{lm} \varepsilon_{lim}}{s + \mu_{limD} + \varepsilon_{lim}}\right)}}{2\dfrac{R_m \alpha_r^2 D_l}{AB^2 R_{lm}}}$$ |
| $E$ | $$s + \varepsilon_m + \mu_{mD} - \frac{\varepsilon_m \varepsilon_{im}}{s + \mu_{imD} + \varepsilon_{im}} + \frac{\theta_{um} \alpha_r^2 v_{um}}{2A\theta_m B} - \frac{\theta_{lm} \alpha_r^2 v_{lm}}{2A\theta_m B} - \frac{a_2 \theta_{um} \alpha_r^2 D_u}{2A\theta_m B^2}$$ $$+ \frac{b_1 \theta_{lm} \alpha_r^2 D_l}{2AB^2 \theta_m}$$ |
| $y_{inj}$ | $r_D + \dfrac{1}{4E}$ |
| $y_{inj,w}$ | $r_{wD} + \dfrac{1}{4E}$ |
| $\varepsilon_m$ | $\dfrac{\omega_a \alpha_r^2}{A\theta_m}$ |
| $\varepsilon_{im}$ | $\dfrac{\omega_a \alpha_r^2}{A\theta_{im}}$ |
| $\varepsilon_{um}$ | $\dfrac{\omega_u \alpha_r^2 R_m}{A\theta_{um} R_{um}}$ |
| $\varepsilon_{uim}$ | $\dfrac{\omega_l \alpha_r^2 R_m}{A\theta_{um} R_{uim}}$ |
| $\beta_{inj}$ | $\dfrac{V_{w,inj} r_{wD}}{\xi R_m \alpha_r}$ |
| $\xi$ | $4\pi r_w \theta B$ |
| $\phi_1$ | $\dfrac{1}{s(s\beta_{inj} + 1)} \dfrac{1}{\exp\left(\dfrac{y_{inj,w}}{2}\right)\left[\dfrac{A_i\left(E^{1/3} y_{inj,w}\right)}{2} - E^{1/3} A_i'\left(E^{1/3} y_{inj}\right)\right]}$ |



**Table 2: Expressions of the coefficients in the solutions expressed in Eqs.(26a) - (26g).**

| | |
|---|---|
| $\delta_1$ | $-\dfrac{\beta_{cha,D}}{s\beta_{cha,D}+1}\dfrac{r_D\|_{r_D\to\infty}}{(r_{wD}-r_D\|_{r_D\to\infty}-1)}C_{inj,mD}(r_D,t_D)\big\|_{t_D=t_{inj,D}}$ |
| $\delta_2$ | $\dfrac{\beta_{cha,D}}{s\beta_{cha,D}+1}\dfrac{1}{(r_{wD}-r_D\|_{r_D\to\infty}-1)}C_{inj,mD}(r_D,t_D)\big\|_{t_D=t_{inj,D}}$ |
| $s_1$ | $-s_2 z_{eD}$ |
| $s_2$ | $\dfrac{\bar{C}_{mD}(r_D,s)}{1-z_{eD}}$ |
| $\beta_{cha,D}$ | $-\dfrac{V_{w,cha}r_{wD}}{\xi R_m \alpha_r}$ |
| $E_a$ | $s+\varepsilon_m+\mu_{mD}-\dfrac{\varepsilon_m\varepsilon_{im}}{s+\mu_{imD}+\varepsilon_{im}}+\dfrac{\theta_{um}\alpha_r^2 v_{um}}{2A\theta_m B}-\dfrac{\theta_{lm}\alpha_r^2 v_{lm}}{2AB^2\theta_m}$ $-\dfrac{1}{1-z_{eD}}\dfrac{\theta_{um}\alpha_r^2 D_u}{2A\theta_m B^2}+\dfrac{1}{z_{eD}-1}\dfrac{\theta_{lm}\alpha_r^2 D_l}{2AB^2\theta_m}$ |
| $E_u$ | $s+\varepsilon_{um}+\mu_{umD}-\dfrac{\varepsilon_{um}\varepsilon_{uim}}{s+\varepsilon_{uim}+\mu_{uimD}}$ |
| $E_l$ | $s+\varepsilon_{lm}+\mu_{lmD}-\dfrac{\varepsilon_{lm}\varepsilon_{lim}}{s+\varepsilon_{lim}+\mu_{limD}}$ |
| $F$ | $C_{mD}(r_D,t_{inj,D})+\dfrac{\varepsilon_m C_{imD}(r_D,t_{inj})}{s+\mu_{imD}+\varepsilon_{im}}$ |
| $\varphi(\eta)$ | $F\eta-[\delta_2+\eta E_a(\delta_1+\delta_2\eta)]$ |
| $f_u(\eta_u)$ | $C_{umD}(r_D,\eta_u,t_{inj,D})+\dfrac{\varepsilon_{um}C_{uimD}(r_D,\eta_u,t_{inj,D})}{s+\varepsilon_{uim}+\mu_{uimD}}-\dfrac{R_m v_{um}\alpha_r^2}{ABR_{um}}s_2$ $-E_u(s_1+s_2\eta_u)$ |
| $f_l(\eta_l)$ | $C_{lmD}(r_D,\eta_l,t_{inj,D})+\dfrac{\varepsilon_{lm}C_{limD}(r_D,\eta_l,t_{inj,D})}{s+\varepsilon_{lim}+\mu_{limD}}+\dfrac{R_m v_{lm}\alpha_r^2}{ABR_{lm}}\dfrac{\bar{C}_{mD}}{z_{eD}-1}$ $-\bar{C}_{mD}E_l\dfrac{z_{eD}+\eta_l}{z_{eD}-1}$ |
| $g(r_D,E_a;\eta)$ | $g_1(r_D,E_a;\eta)=\mathcal{T}_1 exp(\frac{y_{cha}}{2})A_i\left(E_a^{\frac{1}{3}}y_{cha}\right)+\mathcal{T}_2 exp\left(\frac{y_{cha}}{2}\right)B_i\left(E_a^{\frac{1}{3}}y_{cha}\right) r_{wD}\le$ $g_2(r_D,E_a;\eta)=\mathcal{T}_3 exp(\frac{y_{cha}}{2})A_i\left(E_a^{\frac{1}{3}}y_{cha}\right)+\mathcal{T}_4 exp\left(\frac{y_{cha}}{2}\right)B_i\left(E_a^{\frac{1}{3}}y_{cha}\right) \eta\le y$ |
| $g_u(z_D,E_u;\eta_u)$ | $g_{u1}(z_D,E_u;\eta_u)=N_1 exp(a_1 z_D)+N_2 exp(a_2 z_D)\ \ 1\le z_D<\eta_u$ $g_{u2}(z_D,E_u;\eta_u)=N_3 exp(a_1 z_D)+N_4 exp(a_2 z_D)\ \ \eta_u\le z_D<\infty$ |



| $g_l(z_D, E_l; \eta_l)$ | $g_{u1}(z_D, E_l; \eta_l) = M_1 exp(b_1 z_D) + M_2 exp(b_2 z_D) \quad -1 \leq z_D < \eta_l$ <br> $g_{u2}(z_D, E_l; \eta_l) = M_3 exp(b_1 z_D) + M_4 exp(b_2 z_D) \quad \eta_l \leq z_D < -\infty$ |
|---|---|
| $M_1$ | $-M_2 exp(b_1 - b_2)$ |
| $M_2$ | $\dfrac{-AB^2 R_{lm}}{R_m \alpha_r^2 D_l[exp(b_2\eta_l - b_1\eta_l) - b_2 exp(b_2\eta_l)]}$ |
| $M_3$ | $M_2 exp(b_2\eta_l - b_1\eta_l) - M_2 exp(b_1 - b_2)$ |
| $M_4$ | $0$ |
| $N_1$ | $-N_2 exp(a_2 - a_1)$ |
| $N_2$ | $\dfrac{-AB^2 R_{um}}{R_m \alpha_r^2 D_u[(a_1 - a_2)exp(a_2 - a_1)exp(a_1\eta_u)]}$ |
| $N_3$ | $0$ |
| $N_4$ | $N_2 - N_2 exp(a_2 - a_1)exp(a_1\eta_u - a_2\eta_u)$ |
| $X$ | $\dfrac{\frac{1}{2}B_i(E_a^{1/3}y_{cha,w}) - E_a^{1/3}B_i'(E_a^{1/3}y_{cha,w})}{\frac{1}{2}A_i(E_a^{1/3}y_{cha,w}) - E_a^{1/3}A_i'(E_a^{1/3}y_{cha,w})}$ |
| $\mathcal{T}_1$ | $-\dfrac{\pi A_i(y_{ext}|_{r_D=\eta^+})}{E^{1/3}}X$ |
| $\mathcal{T}_2$ | $\dfrac{\pi A_i(y_{ext}|_{r_D=\eta^+})}{E_a^{1/3}}$ |
| $\mathcal{T}_3$ | $\dfrac{\pi A_i(y_{ext}|_{r_D=\eta^+})}{E_a^{1/3}}\left[\dfrac{B_i(y_{ext}|_{r_D=\eta^+})}{A_i(y_{ext}|_{r_D=\eta^+})} - X\right]$ |
| $\mathcal{T}_4$ | $0$ |
| $y_{cha}$ | $r_D + \dfrac{1}{4E_a}$ |
| $y_{cha,w}$ | $r_{wD} + \dfrac{1}{4E_a}$ |





**Table 3: Expressions of the coefficients in the solutions expressed in Eqs.(28a) - (28g).**

| | |
|---|---|
| $\Lambda$ | $C_{mD}(r_D, t_{res}) + \dfrac{\varepsilon_m C_{imD}(r_D, t_{res})}{s + \mu_{imD} + \varepsilon_{im}}$ |
| $\beta_{ext,D}$ | $-\dfrac{V_{w,ext} r_{wD}}{\xi R_m \alpha_r}$ |
| $\zeta$ | $s + \varepsilon_m + \mu_{mD} - \dfrac{\varepsilon_{im}\varepsilon_m}{s + \mu_{imD} + \varepsilon_{im}} - \dfrac{\theta_{um}\alpha_r^2 v_{um}}{2A\theta_m B} + \dfrac{\theta_{lm}\alpha_r^2 v_{lm}}{2AB^2\theta_m}$ $-\dfrac{1}{1-z_{eD}}\dfrac{\theta_{um}\alpha_r^2 D_u}{2A\theta_m b} + \dfrac{1}{z_{eD}-1}\dfrac{\theta_{lm}\alpha_r^2 D_l}{2Ab^2\theta_m}$ |
| $f(\varepsilon)$ | $exp(\varepsilon/2)\varepsilon\Lambda - \left(\varepsilon\zeta + \dfrac{1}{4}\right)(\sigma_1 + \sigma_2\varepsilon)$ |
| $f_u(\ell_u)$ | $C_{umD}(r_D, \ell_u, t_{res,D}) + \dfrac{\varepsilon_{um} C_{uimD}(r_D, \ell_u, t_{res,D})}{s + \varepsilon_{uim} + \mu_{uimD}} + \dfrac{R_m v_{um}\alpha_r^2}{ABR_{um}}\dfrac{\bar{C}_{mD}(r_D, s)}{1-z_{eD}}$ $-\dfrac{\ell_u - z_{eD}}{1-z_{eD}}E_u\bar{C}_{mD}(r_D, s)$ |
| $f_l(\ell_l)$ | $C_{mD}(r_D, \ell_l, t_{res,D}) + \dfrac{\varepsilon_{lm} C_{limD}(r_D, \ell_l, t_{res,D})}{s + \varepsilon_{lim} + \mu_{limD}} - \dfrac{R_m v_{lm}\alpha_r^2}{ABR_{lm}}\dfrac{\bar{C}_{mD}(r_D, s)}{z_{eD}-1}$ $-\dfrac{\ell_l + z_{eD}}{z_{eD}-1}E_l\bar{C}_{mD}(r_D, s)$ |
| $g(r_D, \zeta; \varepsilon)$ | $g_1(r_D, \zeta; \varepsilon) = P_1 A_i(y_{ext}) + P_2 B_i(y_{ext})$ $\quad r_{wD} \leq y_{ext} < \varepsilon$ $g_2(r_D, \zeta; \varepsilon) = P_3 A_i(y_{ext}) + P_4 B_i(y_{ext})$ $\quad\quad \varepsilon \leq y_{ext} < \infty$ |
| $g_u(z_D, E_u; \ell_u)$ | $g_{u1}(z_D, E_u; \ell_u) = H_1 exp(m_1 z_D) + H_2 exp(m_2 z_D)$ $\quad 1 \leq z_D < \ell_u$ $g_{u2}(z_D, E_u; \ell_u) = H_3 exp(m_1 z_D) + H_4 exp(m_2 z_D)$ $\quad \ell_u \leq z_D < \infty$ |
| $g_l(z_D, E_l; \ell_l)$ | $g_{l1}(z_D, E_l; \ell_l) = I_1 exp(n_1 z_D) + I_2 exp(n_2 z_D)$ $\quad -1 \leq z_D < \ell_l$ $g_{l2}(z_D, E_l; \ell_l) = I_3 exp(n_1 z_D) + I_4 exp(n_2 z_D)$ $\quad \ell_l \leq z_D < -\infty$ |
| $H_1$ | $-H_2 exp(m_2 - m_1)$ |
| $H_2$ | $\dfrac{-AR_{um}B^2}{R_m\alpha_r^2 D_u[(m_1 - m_2)exp(m_2 - m_1)exp(m_1\ell_u)]}$ |
| $H_3$ | $0$ |
| $H_4$ | $H_2 - H_2 exp(m_2 - m_1)exp(m_1\ell_u - m_2\ell_u)$ |
| $I_1$ | $-I_2 exp(n_1 - n_2)$ |
| $I_2$ | $\dfrac{-AB^2 R_{lm}}{R_m\alpha_r^2 D_l[exp(n_2\ell_l - n_1\ell_l) - n_2 exp(n_2\ell_l)]}$ |
| $I_3$ | $I_2 exp(n_2\ell_l - n_1\ell_l) - I_2 exp(n_1 - n_2)$ |





| | |
|---|---|
| $I_4$ | $0$ |
| $m_1$ | $\dfrac{-\dfrac{R_m v_{um}\alpha_r^2}{ABR_{um}} + \sqrt{\left(\dfrac{R_m v_{um}\alpha_r^2}{ABR_{um}}\right)^2 + 4\dfrac{R_m \alpha_r^2 D_u}{AB^2 R_{um}}\left(s + \varepsilon_{um} + \mu_{umD} - \dfrac{\varepsilon_{um}\varepsilon_{uim}}{s + \mu_{uimD} + \varepsilon_{uim}}\right)}}{2\dfrac{R_m \alpha_r^2 D_u}{AB^2 R_{um}}}$ |
| $m_2$ | $\dfrac{-\dfrac{R_m v_{um}\alpha_r^2}{ABR_{um}} - \sqrt{\left(\dfrac{R_m v_{um}\alpha_r^2}{ABR_{um}}\right)^2 + 4\dfrac{R_m \alpha_r^2 D_u}{AB^2 R_{um}}\left(s + \varepsilon_{um} + \mu_{umD} - \dfrac{\varepsilon_{um}\varepsilon_{uim}}{s + \mu_{uimD} + \varepsilon_{uim}}\right)}}{2\dfrac{R_m \alpha_r^2 D_u}{AB^2 R_{um}}}$ |
| $n_1$ | $\dfrac{\dfrac{R_m v_{lm}\alpha_r^2}{ABR_{lm}} + \sqrt{\left(\dfrac{R_m v_{lm}\alpha_r^2}{ABR_{lm}}\right)^2 + 4\dfrac{R_m \alpha_r^2 D_l}{AB^2 R_{lm}}\left(s + \varepsilon_{lm} + \mu_{lmD} - \dfrac{\varepsilon_{lm}\varepsilon_{lim}}{s + \mu_{limD} + \varepsilon_{lim}}\right)}}{2\dfrac{R_m \alpha_r^2 D_l}{AB^2 R_{lm}}}$ |
| $n_2$ | $\dfrac{\dfrac{R_m v_{lm}\alpha_r^2}{ABR_{lm}} - \sqrt{\left(\dfrac{R_m v_{lm}\alpha_r^2}{ABR_{lm}}\right)^2 + 4\dfrac{R_m \alpha_r^2 D_l}{AB^2 R_{lm}}\left(s + \varepsilon_{lm} + \mu_{lmD} - \dfrac{\varepsilon_{lm}\varepsilon_{lim}}{s + \mu_{limD} + \varepsilon_{lim}}\right)}}{2\dfrac{R_m \alpha_r^2 D_l}{AB^2 R_{lm}}}$ |
| $P_1$ | $-\dfrac{\pi A_i\left(y_{ext}\big|_{r_D=\varepsilon^+}\right)}{\zeta^{1/3}}W$ |
| $P_2$ | $\dfrac{\pi A_i\left(y_{ext}\big|_{r_D=\varepsilon^+}\right)}{\zeta^{1/3}}$ |
| $P_3$ | $\dfrac{\pi A_i\left(y_{ext}\big|_{r_D=\varepsilon^+}\right)}{\zeta^{1/3}}\left[\dfrac{B_i\left(y_{ext}\big|_{r_D=\varepsilon^+}\right)}{A_i\left(y_{ext}\big|_{r_D=\varepsilon^+}\right)} - W\right]$ |
| $P_4$ | $0$ |
| $W$ | $\dfrac{\left(s\beta_{ext,D} + \dfrac{1}{2}\right)B_i\left(y_{ext,w}\right) - \zeta^{1/3}B_i'\left(y_{ext,w}\right)}{\left(s\beta_{ext,D} + \dfrac{1}{2}\right)A_i\left(y_{ext,w}\right) - \zeta^{1/3}A_i'\left(y_{ext,w}\right)}$ |
| $y_{ext}$ | $\zeta^{1/3}\left(r_D + \dfrac{1}{4\zeta}\right)$ |
| $y_{ext,w}$ | $\zeta^{1/3}\left(r_{wD} + \dfrac{1}{4\zeta}\right)$ |
| $\sigma_1$ | $-\dfrac{\beta_{ext,D}\exp(r_{wD}/2)C_{mD}\left(r_{wD}, t_{res,D}\right)}{\left(s\beta_{ext,D} + \dfrac{1}{2}\right)r_{wD} - 1 - \left(s\beta_{ext,D} + \dfrac{1}{2}\right)r_D\big|_{r_D\to\infty}}r_D\big|_{r_D\to\infty}$ |





| $\sigma_2$ | $\dfrac{\beta_{ext,D}\, exp(r_{wD}/2)\, C_{mD}\left(r_{wD}, t_{res,D}\right)}{\left(s\beta_{ext,D}+\frac{1}{2}\right)r_{wD}-1-\left(s\beta_{ext,D}+\frac{1}{2}\right)r_D\big|_{r_D\to\infty}}$ |
|---|---|


**Table 4: A partial list of parameters from literatures.**

|  | Fine sand | Medium sand | Course sand | Clay |
|---|---|---|---|---|
| Retardation factor [-] | 1.20-4.76[a] | 11.40-13.24[b] | 1.10-7.30[c] | 6.98[d] |
| Dispersivity [cm] | 0.15-0.21[e] | 0.20-9.00[b] | 3.2-38.6[c] | 13.80[f] |
| First-order mass transfer coefficient[1/d] | 0.15-0.40[g] | 0.50[g] | 1.0-4.6[g] | 0.05-0.15[g] |
| Porosity [-] | 0.28-0.31[e] | 0.36[b] | 0.37-0.40[e] | 0.40-0.44[f] |
| Reaction rate[1/d] | 6.36-6.84[h] | 0.08-2.1[i] | 0.55-3.12[j] | 0.10-28.80[k] |

[a]. Brusseau et al. (1991); [b]. Pickens et al. (1981); [c].Davis et al. (2003); [d].Javadi et al. (2017); [e].Liang et al. (2018) ;[f].Swami et al. (2016); [g].Kookana et al. (1992); [h].Haggerty et al. (1998); [i].Bouwer and McCarty (1985); [j].Chun et al. (2009); [k].Alvarez et al. (1991). References are shown in Section S3 of *Supplementary Materials*.
