# Peer review of "New Model of Reactive Transport in Single-Well Push-Pull Test with Aquitard Effect and Wellbore Storage"

_Hydrology and Earth System Sciences, 2019_

## Referee Comment (RC1) · Anonymous Referee #1 · 4 Mar 2020

This is an impressive mathematical work that involves several injection phases, adsorption (linear) and first-order degradation, the presence of aquitards, and the separation between mobile/immobile domains. The solution is fully analytical, just expressed in Laplace space (thus the need for inversion at the end). If the solution is analytical, what is the point to test it? The only reason is that some simplifications are involved. This is tested for example in Figure 2, showing limitations.

Assumptions are quite strong: - Homogeneity – it might also be valid for mild heterogeneity - The well extends all the thickness of the aquifer - Reactions: actually you only include linear sorption ($K_d$ values) and first-order degradation ($\mu$ values). This is a very small subset of reactions.

At the end there is a validation effort with real data. According to the authors, the new

model performs better. Yes, it also has many more parameters, and so in a real case some model selection criteria should be performed to discriminate the "best" model. More, the authors provide just a single set of parameters, without any study of uncertainty in the parameters, or even the reason why these numbers were chosen and how they represent real physical quantities.

The mathematical work is really impressive, and I praise the authors for it, but in my opinion the resulting work can be hardly used with real data, and the problem would be better solved using a numerical model that can provide best fit, but also some uncertainty evaluation.

---

## Referee Comment (RC2) · Anonymous Referee #2 · 17 Mar 2020

The present work presents a novel analytical treatment of single-well injection withdrawal (SWIW) tests, whereas the impact of mixing in the well and the presence of confining aquitards are considered. I applaud the Authors for the efforts in the derivation of the solution (math not checked) and the commitment to introduce more flexibility in the conceptual model. Yet, I am not sure about its usefulness to other researchers, it is very complicated! Maybe, if the Authors made available a script for the calibration against data it could be beneficial to the usage among practitioners. Regarding the quality of the paper, I see many unclear points or unclear parts. I listed below a series of comments which I hope will made the paper more clear. Moreover, I have some criticisms about the employed sensitivity analysis, which it seems to be a weak one in my personal opinion.

Comment 1: line 15, why put emphasis on the use of Green' function for the extraction phase in the abstract? This leads to think 'what about the other phases?'. I would remove this comment.

Comment 2: line 17: I would replace 'tested by' with 'tested against results grounded on numerical simulations', or something similar, i.e., the numerical simulations results serve as reference values to be matched and do not verify the validity of the assumptions directly.

Comment 3: lines 17-19 'The sensitivity analysis demonstrates that the influence of vertical flow velocity and porosity in the aquitards, and radial dispersion of the aquifer is more sensitive to the SWIW test than other parameters.'. Which sensitivity analysis? The fact that the latter has been conducted is not specify earlier in the text. Moreover, specify which kind of sensitivity analysis you are using. Furthermore, the sentence is rather confusing: it says that the influence of three parameter is more sensitive to the SWIW test, than other. What is the difference between influence and sensitivity? Is it the influence that varies as a function of the SWIW test? ... I was thinking that are the results of the SWIW (i.e., model output) to be largely sensitive (i.e., influenced by) to the three mentioned parameters (i.e., model inputs), but maybe I am biased by my previous experiences with sensitivity analysis. Please clarify.

Comment 4: line 23 'The new model of this study performs better than previous studies excluding the aquitard effect for interpreting data of the field SWIW test' too general. Please specify which field test you are referring to, since the quality of the novel solution can be worst than previous ones in case the system do not have an aquitard, for example.

Comment 5: lines 49-50 'Another assumption included in many previous models of radial dispersion is that the concentration of the mixing water with the injected tracer is equal to the injected tracer concentration during the injection phase' the sentence is not very clear. What is the mixing water? 'is equal to the injected tracer concentration'

of what? Please revise the sentence. Moreover, lines 53-55 'This assumption implies that the mixing effect in the wellbore is not considered, where the mixing effect refers to the mixture between the original (or native) water and the injected tracer in the well.' Now there is the native water which is not mentioned earlier. ... I can grasp the general idea that there is a difference between the concentration of tracer between the resident water, injected water and water within the well where mixing occurs, but not in a stand-alone manner from these lines (i.e., I need to think about them and deduce that this the implied message). Please clarify, maybe with an additional figure.

Comment 6: line 61 'mostly because ADE could not adequately interpret anomalous reactive transport,' this true when the ADE is used to capture the whole behavior of the system, i.e., as an effective model for all the system behavior to be characterized by a single representative value of advection, dispersion and reaction. Instead, if ADE is finely discretized (i.e., the system heterogeneity is properly detailed) and then (numerically) solved it can fairly well capture anomalous behaviors. Please clarify this point. This is in line with the mentioned superior capacity of effective transport models mentioned afterward (e.g., MMT,CTRW, fADE, MIM) to have a superior capacity in rendering anomalous behaviors of heterogeneous system when viewed as a whole (e.g., spatially integrated BTCs).

Comment 7: line 74 'anonymous' I suppose anomalous.

Comment 8: line 86 'Some examples of weak heterogeneity include the Borden Site of Canada (Sudicky, 1988)' this is just one example, either add others or modify the sentence.

Comment 9: lines 89-96 'Second, for moderate or even strong heterogeneous media such as Cape Code site (Hess, 1989) or MADE site (Bohling et al., 2012), the analytical model developed under the homogeneity assumption is also valuable, but in a statistical sense, as long as the media heterogeneity can be regarded as spatially stationary, meaning that the statistical structure of the media heterogeneity does not

vary in space. In this setting, the analytical model developed under the homogeneity assumption is used to describe the (ensemble) average characteristics of an ensemble of heterogeneous media which are statistically identical but individually different. In another word, such an analytical model will provide a statistically average description of many realizations (an ensemble) which are similar to the heterogeneous media of concern, but it cannot provide an exact description for the particular heterogeneous media under investigation' ... this made me think that the validation strategy based on the direct numerical simulations is not valid: those simulations are considering directly an homogenous media (with deterministic properties) and NOT the statistical average of the SWIW results across a set of Monte Carlo realizations of the conductivity fields, characterized by either small, middle or large variance. Please clarify this point.

Comment 10: line 99 'A schematic diagram of the model investigated by this study is similar to Figure 1 of Wang and Zhan (2013)' please add this figure and incorporate what mentioned above in comment 5.

Comment 11: Eq.s (1)-(3) I didn't quite understand the + notation: I would say that the fact that the velocity component is pointing towards the well or in the opposite direction is in the value of (for example) va considering (1), similar for the others velocity components in (2) and (3). I would say that the value of va (and others advective velocities) varies as a function of the SWIW phase. If not va should be the module of the advective component, no? Maybe I am wrong.

Comment 12: Eq. (12a) what's C0? (12d) there is a $\theta$ without subscript, what's that?

Comment 13: Eq.s (8)-(11) Highlight that in the imposition of the continuity of flux across the well and the formation only the mobile fractions are considered, for who are not familiar with the MIM model?

Comment 14: 'For instance, if the characteristic length of SWIW test is l and the aquifer hydraulic diffusivity is D=Ka/Sa, where Ka are Sa are the radial hydraulic conductivity and specific storage, then the typical characteristic time of unsteady state flow is

around tc = l^2/2D. For instance, for a typical lc=10 m, Ka=10 m/day and Sa=10-5 (m-1) (which are representative of an aquifer consisting of medium sands), the value of tc is found to be 5x10^-5 day.' How do the authors determine the characteristic length lc? In my experience this length is typically a function of the aquifer diffusivity, e.g., for tidal fluctuations is idealized coastal aquifer (e.g., homogeneous, infinite lateral extension) there is a proportionality of the kind lc = sqrt(K/S) (see e.g., Guarracino et al., 2012). Moreover, the proposed estimate of 10 m disagree with the results presented in figures 2-3 where the solute travels up to 100 m, suggesting that the influence of the SWIW test is at least reaching that distance. I am not entirely convinced about the fact that push-pull tests can be seen as steady state tests and with the justification provided by the Authors, I leave to the Editor the judgment here. Nevertheless, I agree on the need to simplify the (already complex) analysis choosing the steady state!

Comment 15: line 289, in the comparison against the numerical solution the porosity of the immobile region of the aquifer is zero, why? There is also a general $\omega=0$, to which mass transfer makes it reference? Why zero? Aren't these choices limiting the testing of the proposed solution?

Comment 16: lines 309-310 'As mentioned in Section 3.1, the new model is a generalization of many previous models, and the conceptual model is more close to reality.' Again, too general. This novel solution could or not be closer to reality depending on the specific case.

Comment 17: line 323 'To prioritize the sensitivity of parameters involved the new model' an in is missing (i.e., 'in the new model'). Moreover, the sensitivity is not a property of the parameters (or model inputs), but it is of the output with respect to the parameters. You want to quantify/evaluate the sensitivity of predictions with respect to the diverse parameters. Sensitivity cannot be prioritized, it is what it is and it is dictated by the way a model builds relationship between input(s) and output(s). Then you can prioritize the estimate of those parameters that influence the most the output.

[Figure]

Comment 18: Eq. (19), the definition and explanation is quite obscure. The only clar thing is that it sensitivity is grounded here on the concept of derivative. Then, what is ci? Moreover, the subscript i does not vary at all, what is it? Why there is Ij before the derivative?. Furthermore, this equation implies (i) that only variation of a single parameter at time are considered and (ii) it seems that the index associated with a parameter is evaluated around only one value of that parameter. These features prevent the identification of non-linearities and parameters interactions, which are quite likely to occur for the present model. The proposed method is a quite restricted characterization of sensitivity to me, if the model is not expensive I would suggest to use a global sensitivity method: Sobol' indices (see Sobol, 2001) or DELSA (see Rakovec et al., 2014). On this point I leave the final decision to the Editor.

Comment 19: lines 389-390 'The new model is most sensitive to the aquitard porosity and aquifer radial dispersivity' the model results are ... 'after a comprehensive sensitivity analysis' you discover the previous thing after performing the sensitivity analysis, and it is not the latter that implies the former results; the sensitivity analysis is just a way to quantify the former aspect. Moreover, I would avoid comprehensive, see comment 18.

Reference

Guarracino, L., Carrera, J., Vázquez-Suñé, E. (2012), Analytical study of hydraulic and mechanical effects on tide-induced head fluctuation in a coastal aquifer system that extends under the sea, J. Hydrol., 450-451, 150-158, https://doi.org/10.1016/j.jhydrol.2012.05.015.

Rakovec, O., M. C. Hill, M. P. Clark, A. H. Weerts, A. J. Teuling, and R. Uijlenhoet (2014), Distributed Evaluation of LocalSensitivity Analysis (DELSA), with application to hydrologic models,Water Resour. Res.,50, 409–426, doi:10.1002/2013WR014063.

Sobol', I. M. (2001), Global sensitivity indices for nonlinear mathematicalmodels and their Monte Carlo estimates,Math. Comput. Simul.,55(1–3),271–280,

doi:10.1016/S0378-4754(00)00270-6.

---

## Author Comment (AC1) · 8 May 2020

The comment was uploaded in the form of a supplement:
https://www.hydrol-earth-syst-sci-discuss.net/hess-2019-699/hess-2019-699-AC1-supplement.pdf

---

## Author Comment (AC2) · 8 May 2020

**TEXAS A&M UNIVERSITY**
DEPARTMENT OF GEOLOGY & GEOPHYSICS
COLLEGE STATION, TEXAS 77843-3115

Dr. Hongbin Zhan, Endowed Dudley J. Hughes Chair in Geology and Geophysics
Tele: (979) 862-7961   Fax (979) 845-6162
Email:zhan@geos.tamu.edu
http://geoweb.tamu.edu/zhan

May 3, 2020

Memorandum

To: Dr. Philippe Ackerer, Editor of HESS

Subject: Revision of Paper # hess-2019-699
* * *
Dear Editor:

Upon the recommendation, we have carefully revised Paper # hess-2019-699 entitled "New Model of Reactive Transport in Single-Well Injection-Withdrawal Test with Aquitard Effect" after considering all the comments made by the reviewers. The following is the point-point response to all the comments.

**Response to Reviewer #1:**

**General comments**

1. This is an impressive mathematical work that involves several injection phases, adsorption (linear) and first-order degradation, the presence of aquitards, and the separation between mobile/immobile domains. The solution is fully analytical, just expressed in Laplace space (thus the need for inversion at the end). If the solution is analytical, what is the point to test it? The only reason is that some simplifications are involved. This is tested for example in Figure 2, showing limitations.

**Reply:** Implemented. See Lines 282-286.
The newly derived analytical solutions will be tested for two aspects, as shown in Section 4. Firstly, the new solution of this study could reduce to the previous solutions under special cases, as the model established in this study is an extension of the previous ones, and the comparisons between them will be shown in Section 4.1. Secondly, although some assumptions included in the previous models have been relaxed in the new model, some other processes of the reactive transport in the SWPP test have to be simplified in the analytical solutions. Assumptions included in the new model have been discussed and their applicability is elaborated in Section 4.2.

2. Assumptions are quite strong: - Homogeneity – it might also be valid for mild heterogeneity - The well extends all the thickness of the aquifer - Reactions: actually you only include linear sorption (K_d values) and first-order degradation (nmu values). This is a very small subset of reactions.

**Reply:** Implemented. See Lines 91-95.

Such assumptions might be oversimplified for most cases in reality, while they are inevitable for the derivation of the analytical solution, especially for the aquifer homogeneity. For a heterogeneity aquifer, the solution presented here may be regarded as an ensemble-averaged approximation if the heterogeneity is spatially stationary. If the heterogeneity is spatially non-stationary, then one can apply non-stationary

stochastic approach and/or Monte Carlo simulations to deal with the issue, which is out of the scope of this investigation.

3. At the end there is a validation effort with real data. According to the authors, the new model performs better. Yes, it also has many more parameters, and so in a real case some model selection criteria should be performed to discriminate the "best" model. More, the authors provide just a single set of parameters, without any study of uncertainty in the parameters, or even the reason why these numbers were chosen and how they represent real physical quantities.

**Reply:** Implemented. The real physical quantities and the uncertainty of the estimated parameters have been discussed. See Lines 383-391.

The values of retardation factor and reaction rate represent that the chemical reaction and sorption are weak for the tracer of KBr in the SWPP test. It is not surprising since KBr is commonly treated as a "conservative" tracer. The porosity of the real aquifer ranges from 0.01 to 0.1, according to the well log analysis (Yang et al., 2014). The estimated porosity represents the average values of the aquifer and aquitards. The estimated dispersivity of the aquifer is 0.7134m by Chen et al. (2017), which is similar with ours. The values of water level in the test could be observed directly; however, these data are not available, and they have to be estimated in this study. To evaluate the uncertainty in the estimated parameters, the sensitivity of the dispersivity on BTCs is analysed, as shown in Figures 8b. One may conclude that the estimated values of this study seem to be representative of the reality.

4. The mathematical work is really impressive, and I praise the authors for it, but in my opinion the resulting work can be hardly used with real data, and the problem would be better solved using a numerical model that can provide best fit, but also some uncertainty evaluation.

**Reply:** Implemented. See Lines 79-88.

The model of the mixing effect will be developed using a mass balance principle in the chaser phase. It seems not difficult to solve this model of this study using the numerical packages, like MODFLOW-MT3DMS, TOUGH and TOUGHREACT, FEFLOW, and so on. However, the numerical solutions may cause errors in modeling the mixing processes in the wellbore, since the volume of the water in the wellbore was assumed to be constant (Wang et al., 2018), while in reality it changes with time. Meanwhile, the numerical errors (like numerical dispersion and numerical oscillation) have to be considered in solving the ADE equation, especially for advection-dominated transport. In this study, analytical solution will be derived to facilitate the data interpretation. Due to the format of analytical solutions, it is much easier to couple such solutions with a proper optimization algorithm (like genetic algorithm). The analytical solution could serve as a benchmark to test the numerical solutions as well.

**Response to Reviewer #2:**

**General comments**

1.  The present work presents a novel analytical treatment of single-well injection withdrawal (SWIW) tests, whereas the impact of mixing in the well and the presence of confining aquitards are considered. I applaud the Authors for the efforts in the derivation of the solution (math not checked) and the commitment to introduce more flexibility in the conceptual model. Yet, I am not sure about its usefulness to other researchers, it is very complicated! Maybe, if the Authors made available a script for the calibration against

data it could be beneficial to the usage among practitioners. Regarding the quality of the paper, I see many unclear points or unclear parts. I listed below a series of comments which I hope will make the paper more clear. Moreover, I have some criticisms about the employed sensitivity analysis, which it seems to be a weak one in my personal opinion.

**Reply:** Thanks. We have carefully revised the manuscript after considering all the comments.

**Specific comments**

1. line 15: why put emphasis on the use of Green' function for the extraction phase in the abstract? This leads to think 'what about the other phases?'. I would remove this comment.

**Reply:** Implemented. We have removed it. See Line 14.

2. line 17: I would replace 'tested by' with 'tested against results grounded on numerical simulations', or something similar, i.e., the numerical simulations results serve as reference values to be matched and do not verify the validity of the assumptions directly.

**Reply:** Implemented.  "tested by a numerical solution" has been changed into "against results grounded on numerical simulations". See Line 16.

3. lines 17-19 'The sensitivity analysis demonstrates that the influence of vertical flow velocity and porosity in the aquitards, and radial dispersion of the aquifer is more sensitive to the SWIW test than other parameters.'. Which sensitivity analysis? The fact that the latter has been conducted is not specify earlier in the text. Moreover, specify which kind of sensitivity analysis you are using. Furthermore, the sentence is rather confusing: it says that the influence of three-parameter is more sensitive to the SWIW test, than other. What is the difference between influence and sensitivity? Is it the influence that varies as a function of the SWIW test? … I was thinking that are the results of the SWIW (i.e., model output) to be largely sensitive (i.e., influenced by) to the three mentioned parameters (i.e., model inputs), but maybe I am biased by my previous experiences with sensitivity analysis. Please clarify.

**Reply:** Implemented. See Lines 17-18.

The results indicate that the results of the SWPP test are mostly sensitive (i.e., influenced by) to the parameters of vertical flow velocity, porosity, and radial dispersion.

4. line 23 'The new model of this study performs better than previous studies excluding the aquitard effect for interpreting data of the field SWIW test' too general. Please specify which field test you are referring to, since the quality of the novel solution can be worse than previous ones in case the system do not have an aquitard, for example.

**Reply:** Implemented. See Line 21, and Lines 271-275.

The new model of this study performs better than previous studies excluding the aquitard effect for interpreting data of the field SWPP test reported by Yang et al. (2014). The new SWPP test model is a generalization of many previous studies; for instance, the new solution reduces to the solution of Gelhar and Collins (1971) when $\omega_a = \omega_u = \omega_l = D_u = D_l = v_{um} = v_{lm} = V_{w,inj} = V_{w,cha} = V_{w,ext} = 0$,

to the solution of Chen et al. (2017) when $\omega_u = \omega_l = D_u = D_l = v_{um} = v_{lm} = V_{w,inj} = V_{w,cha} = V_{w,ext} = 0$, and Wang et al., (2018) when $\omega_a = \omega_u = \omega_l = D_u = D_l = v_{um} = v_{lm} = 0$. Actually, the all values of $\omega_a, \omega_u, \omega_l, D_u, D_l, v_{um}, v_{lm}, V_{w,inj}, V_{w,cha}$, and $V_{w,ext}$ are not zero in the reality, which have been considered in the new solutions of this study.

5. lines 49-50 'Another assumption included in many previous models of radial dispersion is that the concentration of the mixing water with the injected tracer is equal to the injected tracer concentration during the injection phase' the sentence is not very clear. What is the mixing water? 'is equal to the injected tracer concentration' of what? Please revise the sentence. Moreover, lines 53-55 'This assumption implies that the mixing effect in the wellbore is not considered, where the mixing effect refers to the mixture between the original (or native) water and the injected tracer in the well.' ow there is the native water which is not mentioned earlier. … I can grasp the general idea that there is a difference between the concentration of tracer between the resident water, injected water and water within the well where mixing occurs, but not in a standalone manner from these lines (i.e., I need to think about them and deduce that this the implied message). Please clarify, maybe with an additional figure.

**Reply:** Implemented. See Lines 48-68.

Another assumption included in many previous models of radial dispersion is that the mixing effect is ignored. Mixing effect refers to the mixing processes between the prepared tracer injected into the wellbore and original (or native) water in the wellbore in the injection phase of the SWPP test, as shown in Figure 1. As a result of the mixing effect, the concentration inside the wellbore varies with time until reaching the same value as the injected concentration. When ignoring the mixing effect, the concentration inside the wellbore is constant during the entire inject phase, which is certainly not true. The examples of employing a constant wellbore concentration (i.e., the mixing effect is not taken into consideration) include Gelhar and Collins (1971), Chen (1985, 1987), Moench (1989), Chen et al. (2007, 2012), Schroth et al. (2001), Tang and Babu (1979), Chen et al. (2017), Huang et al. (2010), Chen et al. (2012), and Zhou et al. (2017). Recently, Wang et al. (2018) developed a two-phase (injection and extraction) model for the SWPP test with specific considerations of the mixing effect. In many field applications, the chaser and rest phases are generally involved and the mixing effect also happens in these two phases in the SWPP test, which is investigated in this study.

6. line 61 'mostly because ADE could not adequately interpret anomalous reactive transport,' this true when the ADE is used to capture the whole behavior of the system, i.e., as an effective model for all the system behavior to be characterized by a single representative value of advection, dispersion and reaction. Instead, if ADE is finely discretized (i.e., the system heterogeneity is properly detailed) and then (numerically) solved it can fairly well capture anomalous behaviors. Please clarify this point. This is in line with the mentioned superior capacity of effective transport models mentioned afterward (e.g., MMT,CTRW, fADE, MIM) to have a superior capacity in rendering anomalous behaviors of heterogeneous system when viewed as a whole (e.g., spatially integrated BTCs).

**Reply:** Implemented. See Lines 61-75.

Besides above-mentioned issues in previous studies, another issue is that the advection-dispersion equation (ADE) was used to govern the reactive transport of SWPP tests (Gelhar and Collins,1971; Wang et al.; 2018; Jung and Pruess, 2012). The validity of ADE was challenged by numerous laboratory and field experimental studies before, when using a single representative value of advection, dispersion and reaction

to characterize the whole system. In a hypothetical case, if great details of heterogeneity are known, one may employ a sufficiently fine mesh to discretize the porous media of concern and use ADE to capture anomalous transport characteristics fairly well (e.g. the early arrivals and/or heavy late-time tails of the breakthrough curves (BTCs)). However, such a hypothetical case is rarely been materialized in real applications, especially for field-scale problems. To remedy the situation (at least in some degrees), the multi-rate mass transfer (MMT) model was proposed as an alternative to interpret the data of SWPP test (Huang et al., 2010; Chen et al., 2017). In the MMT model, the porous media is divided into many overlapping continuums (Haggerty et al., 2000;Haggerty and Gorelick, 1995). A subset of MMT is the two overlapping continuums or the mobile-immobile model (MIM) in which the mass transfer between two domains (mobile and immobile) becomes a single parameter instead of a function. The MIM model can grasp most characteristics of MMT and is mathematically simpler than MMT. Besides the MMT model, the continuous time random walk (CTRW) model and the fractional advection-dispersion equation (FADE) model were also applied for anomalous reactive transport in SWPP tests (Hansen et al., 2017; Chen et al., 2017). Due to the complexity of the mathematic models of CTRW and FADE, it is very difficult, or even not possible to derive analytical solutions for those two models, although both methods perform well in a numerical framework.

7. line 74 'anonymous' I suppose anomalous.

**Reply:** Implemented. "anonymous" has been changed into "anomalous". See Line 75.

8. line 86 'Some examples of weak heterogeneity include the Borden Site of Canada (Sudicky, 1988)' this is just one example, either add others or modify the sentence.

**Reply:** Implemented. The Borden Site of Canada (Sudicky, 1988) is one example of weak aquifer heterogeneity. See Lines 98-99.

9. lines 89-96 'Second, for moderate or even strong heterogeneous media such as Cape Code site (Hess, 1989) or MADE site (Bohling et al., 2012), the analytical model developed under the homogeneity assumption is also valuable, but in a statistical sense, as long as the media heterogeneity can be regarded as spatially stationary, meaning that the statistical structure of the media heterogeneity does not vary in space. In this setting, the analytical model developed under the homogeneity assumption is used to describe the (ensemble) average characteristics of an ensemble of heterogeneous media which are statistically identical but individually different. In another word, such an analytical model will provide a statistically average description of many realizations (an ensemble) which are similar to the heterogeneous media of concern, but it cannot provide an exact description for the particular heterogeneous media under investigation' .... this made me think that the validation strategy based on the direct numerical simulations is not valid: those simulations are considering directly an homogenous media (with deterministic properties) and NOT the statistical average of the SWIW results across a set of Monte Carlo realizations of the conductivity fields, characterized by either small, middle or large variance. Please clarify this point.

**Reply:** Implemented. See Lines 91-95.

The description of 'Second, for moderate or even strong heterogeneous…" in the original manuscript has been deleted. The clarification of this point could be seen in Lines 91-95:

Such assumptions might be oversimplified for most cases in reality, while they are inevitable for the derivation of the analytical solution, especially for the aquifer homogeneity. For a heterogeneity aquifer, the solution presented here may be regarded as an ensemble-averaged approximation if the heterogeneity is spatially stationary. If the heterogeneity is spatially non-stationary, then one can apply non-stationary stochastic approach and/or Monte Carlo simulations to deal with the issue, which is out of the scope of this investigation.

10. line 99 'A schematic diagram of the model investigated by this study is similar to Figure 1 of Wang and Zhan (2013)' please add this figure and incorporate what mentioned above in comment 5.

**Reply:** Implemented. A new figure has been added, See Figure 1.

[Figure]

**Figure 1: The schematic diagram of the SWPP test.**

11. Eq.s (1)-(3) I didn't quite understand the + notation: I would say that the fact that the velocity component is pointing towards the well or in the opposite direction is in the value of (for example) va considering (1), similar for the others velocity components in (2) and (3). I would say that the value of va (and others advective velocities) varies as a function of the SWIW phase. If not va should be the module of the advective component, no? Maybe I am wrong.

**Reply:** Implemented. Eqs. (1) - (3) have been revised.

12. Eq. (12a) what's C0? (12d) there is a without subscript, what's that?

**Reply:** Implemented. See Lines 155-156.

$$\xi = 2\pi r_w \theta_m 2B, \tag{12d}$$

where $h_{w,inj}$ is the wellbore water depth [L] in the injection phase, $C_0$ is concentration [ML$^{-3}$] of prepared tracer.

13. Eq.s (8)-(11) Highlight that in the imposition of the continuity of flux across the well and the formation only the mobile fractions are considered, for who are not familiar with the MIM model?

**Reply:** Implemented. See Lines 149-150.

Eqs. (8) - (11) indicate that the flux continuity across the interface between well and the formation is only considered for the mobile continuum (or mobile domain).

14. 'For instance, if the characteristic length of SWIW test is l and the aquifer hydraulic diffusivity is D=Ka/Sa, where Ka are Sa are the radial hydraulic conductivity and specific storage, then the typical characteristic time of unsteady state flow is around tc = l^2/2D. For instance, for a typical lc=10 m, Ka=10 m/day and Sa=10-5 (m-1) (which are representative of an aquifer consisting of medium sands), the value of tc is found to be 5x10^-5 day.' How do the authors determine the characteristic length lc? In my experience this length is typically a function of the aquifer diffusivity, e.g., for tidal fluctuations is idealized coastal aquifer (e.g., homogeneous, infinite lateral extension) there is a proportionality of the kind lc = sqrt(K/S) (see e.g., Guarracino et al., 2012). Moreover, the proposed estimate of 10 m disagree with the results presented in figures 2-3 where the solute travels up to 100 m, suggesting that the influence of the SWIW test is at least reaching that distance. I am not entirely convinced about the fact that push-pull tests can be seen as steady state tests and with the justification provided by the Authors, I leave to the Editor the judgment here. Nevertheless, I agree on the need to simplify the (already complex) analysis choosing the steady state!

**Reply:** Implemented. See Lines 178-180.

The use of Eq. (15) implies that quasi-steady state flow can be established very quickly near the injection/pumping well, thus the flow velocity becomes independent of time. This approximation is generally acceptable given the very limited spatial range of influence of most SWPP tests. For instance, if the characteristic length of SWPP test is $l$ and the aquifer hydraulic diffusivity is $D=K_a/S_a$, where $K_a$ are $S_a$ are the radial hydraulic conductivity and specific storage, then the typical characteristic time of unsteady-state flow is around $t_c \approx \frac{l^2}{2D}$. The typical characteristic time refers to the time of the flow changing from transient state to quasi steady state, where the spatial distribution of flow velocity does not change while the drawdown varies with time. Previous studies demonstrated that the characteristic length might be a function of the aquifer hydraulic diffusivity, e.g. $l = \sqrt{K_a/S_a}$ for a homogeneous and infinite lateral extended aquifer (Guarracino et al., 2012), but our numerical simulations show that such model may result in great errors. For instance, for $K_a$=0.1m/day and $S_a$=10$^{-4}$m$^{-1}$ (which are representative of an aquifer consisting of medium sands), numerical simulation shows the typical characteristic length $l$=100m, as shown in Figure S1 in **_Supplementary Materials_**, while $l = \sqrt{K_a/S_a}=\sqrt{1000}$. The parameters of the numerical simulation are the same as ones used in Figure 2 and 3. If the values of the $K_a$, $S_a$, $B$, and $Q$ are given, the typical characteristic length could be calculated by the numerical solution. One has $t_c \approx \frac{l^2}{2D} = 5.0 \times 10^{-3}$ day.

Figure S3 shows the flow is in quasi steady state when time greater than $t_c$, since two curves of $t$ =5.0 × $10^{-3}$ day and $t$ =5.0 × $10^{-3}$ day overlap.

[Figure]

**Figure S2.** Drawdown with distance at the end of the injection phase.

[Figure]

**Figure S3.** Spatial distribution of the flow velocity for different time.

15. line 289, in the comparison against the numerical solution the porosity of the immobile region of the aquifer is zero, why? There is also a general $\omega$ =0, to which mass transfer makes it reference? Why zero? Aren't these choices limiting the testing of the proposed solution?

**Reply:** Implemented. We have revised it: $\theta_{im}$=0.05, and $\omega$=0.01d$^{-1}$. See Line 298.

16. lines 309-310 'As mentioned in Section 3.1, the new model is a generalization of many previous models, and the conceptual model is more close to reality.' Again, too general. This novel solution could or not be closer to reality depending on the specific case.

**Reply:** Implemented. The new SWPP test model is a generalization of many previous studies; for instance, the new solution reduces to the solution of Gelhar and Collins (1971) when $\omega_a = \omega_u = \omega_l = D_u = D_l = v_{um} = v_{lm} = V_{w,inj} = V_{w,cha} = V_{w,ext} = 0$, to the solution of Chen et al. (2017) when $\omega_u = \omega_l = D_u = D_l = v_{um} = v_{lm} = V_{w,inj} = V_{w,cha} = V_{w,ext} = 0$, and Wang et al., (2018) when $\omega_a = \omega_u = \omega_l = D_u = D_l = v_{um} = v_{lm} = 0$. Actually, the all values of $\omega_a$, $\omega_u$, $\omega_l$, $D_u$, $D_l$, $v_{um}$, $v_{lm}$, $V_{w,inj}$, $V_{w,cha}$, and $V_{w,ext}$ are not zero in the reality, which have been considered in the new solutions of this study. See Lines 271-275.

17. line 323 'To prioritize the sensitivity of parameters involved the new model' an in is missing (i.e., 'in the new model'). Moreover, the sensitivity is not a property of the parameters (or model inputs), but it is of the output with respect to the parameters. You want to quantify/evaluate the sensitivity of predictions with respect to the diverse parameters. Sensitivity cannot be prioritized, it is what it is and it is dictated by the

way a model builds relationship between input(s) and output(s). Then you can prioritize the estimate of those parameters that influence the most the output.

**Reply:** Implemented.

"in" has been added. See Line 343.

To prioritize the sensitivity of predictions with respect to the diverse parameters involved in the new model, a sensitivity analysis is conducted in Section 5.2. See Lines 344-345.

18. Eq. (29), the definition and explanation is quite obscure. The only clar thing is that it sensitivity is grounded here on the concept of derivative. Then, what is ci? Moreover, the subscript i does not vary at all, what is it? Why there is Ij before the derivative?. Furthermore, this equation implies (i) that only variation of a single parameter at time are considered and (ii) it seems that the index associated with a parameter is evaluated around only one value of that parameter. These features prevent the identification of non-linearities and parameters interactions, which are quite likely to occur for the present model. The proposed method is a quite restricted characterization of sensitivity to me, if the model is not expensive I would suggest using a global sensitivity method: Sobol' indices (see Sobol, 2001) or DELSA (see Rakovec et al., 2014). On this point I leave the final decision to the Editor.

**Reply:** Implemented. See Lines 353-365.

The model of Eq. (29) in the original manuscript is for the local sensitivity analysis, and it has been deleted. Instead, a global sensitivity analysis is conducted using the model of Morris (1991) to investigate the importance of the input parameters on the output concentration.

19. lines 389-390 'The new model is most sensitive to the aquitard porosity and aquifer radial dispersivity' the model results are… 'after a comprehensive sensitivity analysis' you discover the previous thing after performing the sensitivity analysis, and it is not the latter that implies the former results; the sensitivity analysis is just a way to quantify the former aspect. Moreover, I would avoid comprehensive, see comment 18.

**Reply:** Implemented. See Lines 353-365.

A global sensitivity analysis is conducted using the model of Morris (1991). The description of the sensitivity is also revised.

If you have any further questions about this revision, please contact me.

Sincerely Yours,

Hongbin Zhan, PhD, PG.

Professor and

Holder of Endowed Dudley J. Hughes Chair in Geology and Geophysics

---

## Author Response (AR1)

**TEXAS A&M UNIVERSITY DEPARTMENT OF GEOLOGY & GEOPHYSICS COLLEGE STATION, TEXAS 77843-3115**

Dr. Hongbin Zhan, Endowed Dudley J. Hughes Chair in Geology and Geophysics Tele: (979) 862-7961 Fax (979) 845-6162 Email:zhan@geos.tamu.edu http://geoweb.tamu.edu/zhan

May 14, 2020

Memorandum

To: Dr. Philippe Ackerer, Editor of HESS

Subject: Revision of Paper # hess-2019-699

Dear Editor:

Upon the recommendation, we have carefully revised Paper # hess-2019-699 entitled "New Model of Reactive Transport in Single-Well Injection-Withdrawal Test with Aquitard Effect" after considering all the comments made by the reviewers. The following is the point-point response to all the comments.

**Response to Reviewer #1:**

**General comments**

1. This is an impressive mathematical work that involves several injection phases, adsorption (linear) and first-order degradation, the presence of aquitards, and the separation between mobile/immobile domains. The solution is fully analytical, just expressed in Laplace space (thus the need for inversion at the end). If the solution is analytical, what is the point to test it? The only reason is that some simplifications are involved. This is tested for example in Figure 2, showing limitations.

Reply: Implemented. See Lines 291-296.

2. Assumptions are quite strong: - Homogeneity – it might also be valid for mild heterogeneity - The well extends all the thickness of the aquifer - Reactions: actually you only include linear sorption (K\_d values) and first-order degradation (nmu values). This is a very small subset of reactions.

Reply: Implemented. See Lines 95-99.

3. At the end there is a validation effort with real data. According to the authors, the new model performs better. Yes, it also has many more parameters, and so in a real case some model selection criteria should be performed to discriminate the "best" model. More, the authors provide just a single set of parameters, without any study of uncertainty in the parameters, or even the reason why these numbers were chosen and how they represent real physical quantities.

**Reply**: Implemented. The real physical quantities and the uncertainty of the estimated parameters have been discussed. See Lines 405-413.

4. The mathematical work is really impressive, and I praise the authors for it, but in my opinion the resulting

Teaching Through Research College of Geosciences work can be hardly used with real data, and the problem would be better solved using a numerical model that can provide best fit, but also some uncertainty evaluation.

Reply: Implemented. See Lines 83-91.

**Response to Reviewer #2:**

**General comments**

1. The present work presents a novel analytical treatment of single-well injection withdrawal (SWIW) tests, whereas the impact of mixing in the well and the presence of confining aquitards are considered. I applaud the Authors for the efforts in the derivation of the solution (math not checked) and the commitment to introduce more flexibility in the conceptual model. Yet, I am not sure about its usefulness to other researchers, it is very complicated! Maybe, if the Authors made available a script for the calibration against data it could be beneficial to the usage among practitioners. Regarding the quality of the paper, I see many unclear points or unclear parts. I listed below a series of comments which I hope will make the paper more clear. Moreover, I have some criticisms about the employed sensitivity analysis, which it seems to be a weak one in my personal opinion.

**Reply:** Thanks. We have carefully revised the manuscript after considering all the comments.

**Specific comments**

1. line 15: why put emphasis on the use of Green' function for the extraction phase in the abstract? This leads to think 'what about the other phases?'. I would remove this comment.

Reply: Implemented. We have removed it. See Line 14.

2. line 17: I would replace 'tested by' with 'tested against results grounded on numerical simulations', or something similar, i.e., the numerical simulations results serve as reference values to be matched and do not verify the validity of the assumptions directly.

**Reply:** Implemented. "tested by a numerical solution" has been changed into "against results grounded on numerical simulations". See Lines 18-19.

3. lines 17-19 'The sensitivity analysis demonstrates that the influence of vertical flow velocity and porosity in the aquitards, and radial dispersion of the aquifer is more sensitive to the SWIW test than other parameters.'. Which sensitivity analysis? The fact that the latter has been conducted is not specify earlier in the text. Moreover, specify which kind of sensitivity analysis you are using. Furthermore, the sentence is rather confusing: it says that the influence of three-parameter is more sensitive to the SWIW test, than other. What is the difference between influence and sensitivity? Is it the influence that varies as a function of the SWIW test? ... I was thinking that are the results of the SWIW (i.e., model output) to be largely sensitive (i.e., influenced by) to the three mentioned parameters (i.e., model inputs), but maybe I am biased by my previous experiences with sensitivity analysis. Please clarify.

**Reply:** Implemented. See Lines 19-20.

4. line 23 'The new model of this study performs better than previous studies excluding the aquitard effect for interpreting data of the field SWIW test' too general. Please specify which field test you are referring to, since the quality of the novel solution can be worse than previous ones in case the system do not have an aquitard, for example.

Reply: Implemented. See Lines 24-25, and Lines 277-283.

5. lines 49-50 'Another assumption included in many previous models of radial dispersion is that the concentration of the mixing water with the injected tracer is equal to the injected tracer concentration during the injection phase' the sentence is not very clear. What is the mixing water? 'is equal to the injected tracer concentration' of what? Please revise the sentence. Moreover, lines 53-55 'This assumption implies that the mixing effect in the wellbore is not considered, where the mixing effect refers to the mixture between the original (or native) water and the injected tracer in the well.' ow there is the native water which is not mentioned earlier. ... I can grasp the general idea that there is a difference between the concentration of tracer between the resident water, injected water and water within the well where mixing occurs, but not in a standalone manner from these lines (i.e., I need to think about them and deduce that this the implied message). Please clarify, maybe with an additional figure.

Reply: Implemented. See Lines 51-62.

6. line 61 'mostly because ADE could not adequately interpret anomalous reactive transport,' this true when the ADE is used to capture the whole behavior of the system, i.e., as an effective model for all the system behavior to be characterized by a single representative value of advection, dispersion and reaction. Instead, if ADE is finely discretized (i.e., the system heterogeneity is properly detailed) and then (numerically) solved it can fairly well capture anomalous behaviors. Please clarify this point. This is in line with the mentioned superior capacity of effective transport models mentioned afterward (e.g., MMT,CTRW, fADE, MIM) to have a superior capacity in rendering anomalous behaviors of heterogeneous system when viewed as a whole (e.g., spatially integrated BTCs).

Reply: Implemented. See Lines 63-79.

7. line 74 'anonymous' I suppose anomalous.

**Reply:** Implemented. "anonymous" has been changed into "anomalous". See Line 76.

8. line 86 'Some examples of weak heterogeneity include the Borden Site of Canada (Sudicky, 1988)' this is just one example, either add others or modify the sentence.

**Reply:** Implemented. The Borden Site of Canada (Sudicky, 1988) is one example of weak aquifer heterogeneity. See Lines 104-105.

9. lines 89-96 'Second, for moderate or even strong heterogeneous media such as Cape Code site (Hess, 1989) or MADE site (Bohling et al., 2012), the analytical model developed under the homogeneity assumption is also valuable, but in a statistical sense, as long as the media heterogeneity can be regarded

as spatially stationary, meaning that the statistical structure of the media heterogeneity does not vary in space. In this setting, the analytical model developed under the homogeneity assumption is used to describe the (ensemble) average characteristics of an ensemble of heterogeneous media which are statistically identical but individually different. In another word, such an analytical model will provide a statistically average description of many realizations (an ensemble) which are similar to the heterogeneous media under investigation' .... this made me think that the validation strategy based on the direct numerical simulations is not valid: those simulations are considering directly an homogenous media (with deterministic properties) and NOT the statistical average of the SWIW results across a set of Monte Carlo realizations of the conductivity fields, characterized by either small, middle or large variance. Please clarify this point.

Reply: Implemented. See Lines 95-99.

The description of 'Second, for moderate or even strong heterogeneous..." in the original manuscript has been deleted.

Such assumptions might be oversimplified for cases in reality, while they are inevitable for the derivation of the analytical solution, especially for the aquifer homogeneity. For a heterogeneity aquifer, the solution presented here may be regarded as an ensemble-averaged approximation if the heterogeneity is spatially stationary. If the heterogeneity is spatially non-stationary, then one can apply non-stationary stochastic approach and/or Monte Carlo simulations to deal with the issue, which is out of the scope of this investigation.

10. line 99 'A schematic diagram of the model investigated by this study is similar to Figure 1 of Wang and Zhan (2013)' please add this figure and incorporate what mentioned above in comment 5.

**Reply:** Implemented. A new figure has been added, See Figure 1.

(c) Rest phase

(d) Extraction phase

Figure 1: The schematic diagram of the SWPP test.

11. Eq.s (1)-(3) I didn't quite understand the + notation: I would say that the fact that the velocity component is pointing towards the well or in the opposite direction is in the value of (for example) va considering (1), similar for the others velocity components in (2) and (3). I would say that the value of va (and others advective velocities) varies as a function of the SWIW phase. If not va should be the module of the advective component, no? Maybe I am wrong.

**Reply:** Implemented. Eqs. (1) - (3) have been revised.

12. Eq. (12a) what's CO? (12d) there is a without subscript, what's that?

Reply: Implemented. See Lines 159-161.

 $\xi = 2\pi r_w \theta_m 2B,$ (12d) where  $h_{w,ini}$  is the wellbore water depth [L] in the injection phase,  $C_0$  is concentration [ML-3] of prepared tracer.

13. Eq.s (8)-(11) Highlight that in the imposition of the continuity of flux across the well and the formation only the mobile fractions are considered, for who are not familiar with the MIM model?

Reply: Implemented. See Lines 152-154.

Eqs. (8) - (11) indicate that the flux continuity across the interface between well and the formation is only considered for the mobile continuum (or mobile domain).

14. 'For instance, if the characteristic length of SWIW test is I and the aquifer hydraulic diffusivity is D=Ka/Sa, where Ka are Sa are the radial hydraulic conductivity and specific storage, then the typical characteristic time of unsteady state flow is around tc = I^2/2D. For instance, for a typical Ic=10 m, Ka=10 m/day and Sa=10-5 (m-1) (which are representative of an aquifer consisting of medium sands), the value of tc is found to be 5x10^-5 day.' How do the authors determine the characteristic length Ic? In my experience this length is typically a function of the aquifer diffusivity, e.g., for tidal fluctuations is idealized coastal aquifer (e.g., homogeneous, infinite lateral extension) there is a proportionality of the kind Ic = sqrt(K/S) (see e.g., Guarracino et al., 2012). Moreover, the proposed estimate of 10 m disagree with the results presented in figures 2-3 where the solute travels up to 100 m, suggesting that the influence of the SWIW test is at least reaching that distance. I am not entirely convinced about the fact that push-pull tests can be seen as steady state tests and with the justification provided by the Authors, I leave to the Editor the judgment here. Nevertheless, I agree on the need to simplify the (already complex) analysis choosing the steady state!

Reply: Implemented. See Lines 178-190.

In the comment by reviewer: "In my experience this length is typically a function of the aquifer diffusivity, e.g., for tidal fluctuations is idealized coastal aquifer (e.g., homogeneous, infinite lateral extension) there is a proportionality of the kind Ic = sqrt(K/S) (see e.g., Guarracino et al., 2012)", the formula of computing the characteristic length Ic may be not right, since the dimension of sqrt(K/S) is L/sqrt(T), while the dimension of Ic is L. By checking Guarracino et al. (2012), we found that authors employed "sqrt(K/( $\omega$ S))" to calculate the characteristic dampening distance, where  $\omega$  is tidal angular velocity (T-1).

This approximation is generally acceptable given the very limited spatial range of influence of most SWPP tests. For instance, if the characteristic length of SWPP test is *I* and the aquifer hydraulic diffusivity is  $D=K_{\partial}/S_{\partial}$ , where  $K_{\partial}$  are  $S_{\partial}$  are respectively the radial hydraulic conductivity and specific storage, then the typical characteristic time of unsteady-state flow is around  $t_c \approx \frac{l^2}{2D}$ . The typical characteristic time refers to the time of the flow changing from transient state to quasi-steady state, where the spatial distribution of flow velocity does not change while the drawdown varies with time. This model is similar to the model used to calculate the typical characteristic length of the tide-induced head fluctuation in a coastal aquifer system (Guarracino et al., 2012). For  $K_{\partial}=1m/day$ ,  $S_{\partial}=10^{5}m^{-1}$  and l=10m (which are representative of an aquifer consisting of medium sands), one has  $t_c \approx \frac{l^2}{2D} = 5.0 \times 10^{-3}$  day, which is a very small value. To test the model in computing  $t_c$ , the numerical simulation has been conducted, where the other parameters used in the model are the same as ones used in Figures 2 and 3. Figure S2 shows the flow is in quasi-steady state when time is greater than  $t_c$ , since two curves of  $t = 5.0 \times 10^{-3}$  day and  $t = 10.0 \times 10^{-3}$  day overlap. As for the typical characteristic length, if the values of  $K_a$ ,  $S_a$ , and *B* have been estimated by the pumping tests before the SWPP test, it could be calculated by numerical modelling exercises using different simulation times.

15. line 289, in the comparison against the numerical solution the porosity of the immobile region of the aquifer is zero, why? There is also a general  $\omega = 0$ , to which mass transfer makes it reference? Why zero? Aren't these choices limiting the testing of the proposed solution?

**Reply:** Implemented. We have revised it:  $\theta_{im}$ =0.05, and  $\omega$ =0.01d-1. See Line 308.

16. lines 309-310 'As mentioned in Section 3.1, the new model is a generalization of many previous models, and the conceptual model is more close to reality.' Again, too general. This novel solution could or not be closer to reality depending on the specific case.

Reply: Implemented. See Lines 24-25, and Lines 277-283.

17. line 323 'To prioritize the sensitivity of parameters involved the new model' an in is missing (i.e., 'in the new model'). Moreover, the sensitivity is not a property of the parameters (or model inputs), but it is of the output with respect to the parameters. You want to quantify/evaluate the sensitivity of predictions with respect to the diverse parameters. Sensitivity cannot be prioritized, it is what it is and it is dictated by the way a model builds relationship between input(s) and output(s). Then you can prioritize the estimate of those parameters that influence the most the output.

**Reply: Implemented.**

"in" has been added. See Line 339.

To prioritize the sensitivity of predictions with respect to the diverse parameters involved in the new model, a sensitivity analysis is conducted in Section 5.2. See Lines 354-372.

18. Eq. (29), the definition and explanation is quite obscure. The only clar thing is that it sensitivity is grounded here on the concept of derivative. Then, what is ci? Moreover, the subscript i does not vary at all, what is it? Why there is Ij before the derivative?. Furthermore, this equation implies (i) that only variation of a single parameter at time are considered and (ii) it seems that the index associated with a parameter is evaluated around only one value of that parameter. These features prevent the identification of non-linearities and parameters interactions, which are quite likely to occur for the present model. The proposed method is a quite restricted characterization of sensitivity to me, if the model is not expensive I would suggest using a global sensitivity method: Sobol' indices (see Sobol, 2001) or DELSA (see Rakovec et al., 2014). On this point I leave the final decision to the Editor.

**Reply: Implemented. See Lines 355-372.**

The model of Eq. (29) in the original manuscript is for the local sensitivity analysis, and it has been deleted. Instead, a global sensitivity analysis is conducted using the model of Morris (1991) to investigate the importance of the input parameters on the output concentration.

19. lines 389-390 'The new model is most sensitive to the aquitard porosity and aquifer radial dispersivity' the model results are... 'after a comprehensive sensitivity analysis' you discover the previous thing after performing the sensitivity analysis, and it is not the latter that implies the former results; the sensitivity analysis is just a way to quantify the former aspect. Moreover, I would avoid comprehensive, see comment 18.

**Reply: Implemented. See Lines 354-372.**

A global sensitivity analysis is conducted using the model of Morris (1991). The description of the sensitivity is also revised.

If you have any further questions about this revision, please contact me.

Sincerely Yours,

Hongbin Zhan, PhD, PG.

Professor and

Heybinzhan

Holder of Endowed Dudley J. Hughes Chair in Geology and Geophysics

[revised manuscript text omitted]
 lower aquitard; subscripts (L" m" and "im" refers to parameters in the and immobile domains, respectively;; Cm and Clim are the concentrations [ML-3] of the aquifer; Cum and, Cuim are concentrations [ML-3] of the upper aquitard; Clm and Clim are concentrations [ML-3] of the aquifer; Cum and, Cuim are concentrations [ML-3] of the aquifer; Cum and, Cuim are concentrations [ML-3] of the aquifer; Cum and, Cuim are concentrations [ML-3] of the aquifer; Cum and, Clim and Clim are concentrations [ML-3] of the aquifer; Cum and, Cuim are concentrations [ML-3] of the aquifer; Cum and, Cuim are concentrations [ML-3] of the aquifer; Cum and, Cuim are concentrations [ML-3] of the aquifer; Cum and, Cuim are concentrations [ML-3].
140 time [T]; B is half of the aquifer thickness [L]; r is the radial distance [L]; z represents the vertical distance [L]; rw is the well radius [L]; Dr is aquifer dispersion coefficient [L2T-1]; Du and Dl are vertical dispersion coefficients [L2T-1] of the upper aquitard, respectively; va is-represents the average velocity [LT-1] in the aquifer and va = ua/θm; ua is Darcian velocity [LT-1]; vum and vlm are vertical velocities [LT-1] in the aquitards; µm, µum, µuim, µum, µuim, µuim and µlim are reaction rates; θm, θim, θum, θlm and θlim and θlim are the porosities [dimensionless]; Rm = 1 + pbKd/θm, Rlim = 1 + pbKd/θum, Rlim = 1

[revised manuscript text omitted]
 = 2.5 \text{m}, \ \alpha_u = \alpha_l = 0.5 \text{m}, \ \mu_m = \mu_{im} = \mu_{uim} = \mu_{lim} = \mu_{lim} = 10^{-7} \text{s}^{-1}, \ r_w = 0.5 \text{m}, \ Q_{inj} = Q_{cha} = 50 \text{ m}^3/\text{d}, \ Q_{res} = 0 \text{ m}^3/\text{d}, \ Q_{$ 335  $Q_{ext}$ =-50 m3/d,  $t_{inj}$ =250day,  $t_{cha}$ =50day,  $t_{res}$ =50day, B=10m,  $\theta_m$ =0.325,  $\theta_{im}$ =0.005, and  $\omega$ =00.01-d-1. The comparison of concentration between the analytical and numerical solutions is shown in Figures-2-3 and 34.

340

As the first assumption in Section 3.3 has been elaborated in Section 2.2, the following discussion will only focus on the second and third assumptions. Figures, 2a3(a), 2b-3(b) and 2e-3(c) represent the snapshots of concentration distributions in the aquifer along the r-axis at different times. One may conclude that the curves with smaller  $K_u$  and  $K_l$  values are closer to the analytical solution. This is because aquitards with smaller  $K_u$  and  $K_l$  (when  $K_a K_a$  remains constant) could make flow closer to the horizontal direction (or parallel with the aquitard-aquifer interface) in the aquifer and closer to the vertical direction (or perpendicular with the aquitard-aquifer interface) in the aquitard, according